# Translation inhibitory elements from *Hoxa3* and *Hoxa11* mRNAs use uORFs for translation inhibition

**Fatima Alghoul, Schaeffer Laure, Gilbert Eriani, Franck Martin\***

Institut de Biologie Moléculaire et Cellulaire, "Architecture et Réactivité de l'ARN" CNRS UPR9002, Université de Strasbourg, Strasbourg, France

**Abstract** During embryogenesis, Hox mRNA translation is tightly regulated by a sophisticated molecular mechanism that combines two RNA regulons located in their 5'UTR. First, an internal ribosome entry site (IRES) enables cap-independent translation. The second regulon is a translation inhibitory element or TIE, which ensures concomitant cap-dependent translation inhibition. In this study, we deciphered the molecular mechanisms of mouse *Hoxa3* and *Hoxa11* TIEs. Both TIEs possess an upstream open reading frame (uORF) that is critical to inhibit cap-dependent translation. However, the molecular mechanisms used are different. In *Hoxa3* TIE, we identify an uORF which inhibits cap-dependent translation and we show the requirement of the non-canonical initiation factor eIF2D for this process. The mode of action of *Hoxa11* TIE is different, it also contains an uORF but it is a minimal uORF formed by an uAUG followed immediately by a stop codon, namely a 'start-stop'. The 'start-stop' sequence is species-specific and in mice, is located upstream of a highly stable stem loop structure which stalls the 80S ribosome and thereby inhibits cap-dependent translation of *Hoxa11* main ORF.

**\*For correspondence:**
f.martin@ibmc-cnrs.unistra.fr

**Competing interests:** The authors declare that no competing interests exist.

## Introduction

Gene expression constitutes an indispensable cellular process for which the genetic information encodes a functional product, mainly proteins. This process named translation initiates by a cap-dependent mechanism for most cellular mRNAs. It involves a large number of auxiliary proteins termed eukaryotic initiation factors (eIFs) which are required for the recruitment of the ribosomes on the mRNA (*Hinnebusch, 2014*; *Merrick and Pavitt, 2018*; *Pelletier and Sonenberg, 2019*; *Shirokikh and Preiss, 2018*). To ensure fine-tuning of translation, this step is highly regulated. However, several mRNA subclasses are translated by non-canonical mechanisms. For instance, this is the case for homeobox (Hox) mRNAs. Hox genes encode a family of proteins that constitutes transcription factors. Their main function is to orchestrate specific sequential transcription processes during embryonic development. A wealth of experimental data over the last three decades has led to the identification of many *cis*-regulatory elements that control Hox gene transcriptional patterns, thus giving deeper insights into the expression of Hox mRNAs (*Alexander et al., 2009*). In fact, Hox homologues in *Drosophila*, precisely Antp and Ubx genes in the Hom-C cluster, have been suggested to be under translational control during embryonic development (*Oh et al., 1992*). A subgroup of mRNAs produced from the Antp and Ubx loci contain functional internal ribosome entry sites (IRES) that allow their translation using a cap-independent mechanism during development (*Ye et al., 1997*). More recently, the presence of other IRES elements in the 5'UTR of subsets of mice HoxA mRNAs (*Hoxa3, Hoxa4, Hoxa5, Hoxa9,* and *Hoxa11*) has been demonstrated (*Leppek et al., 2020*; *Xue et al., 2015*). Some of these IRES require the presence of the ribosomal protein RpL38 in the ribosome to efficiently initiate translation, thereby explaining the tissue patterning defective phenotype observed with RpL38 knockout mouse (*Kondrashov et al., 2011*). The IRES

activity in these Hox mRNAs was found to be critical for their appropriate expression. Upon the discovery of the IRES elements in Hox mRNAs, other RNA regulons termed translation inhibitory elements (TIEs) were also found in the 5'UTR of Hox mRNAs (*Xue et al., 2015*). TIEs are located upstream of the previously described IRES. According to the study by *Xue et al., 2015*, these elements efficiently inhibit canonical cap-dependent translation in subsets of HoxA mRNAs (*Hoxa3, Hoxa4, Hoxa9*, and *Hoxa11*) by an unknown mechanism (*Xue et al., 2015*). The action of TIE that ensures efficient blockage of cap-dependent translation promotes IRES-mediated cap-independent translation. TIE and IRES act in synergy to ensure tightly regulated translation during organismal development. Indeed, it has been shown that Hox TIEs ensure that Hox mRNAs are translated solely by their IRES element. Thereby, TIEs represent the first example of specific RNA elements dedicated to inhibiting specifically cap-dependent translation in Hox mRNAs. However, the mechanism of action of these elements as well as their structural characterization are still unknown. In this study, we investigate the functional mode of action of TIEs from *Hoxa3* and *Hoxa11* mRNAs that were previously identified and characterized by *Xue et al., 2015*. Our study was conducted on *Hoxa3* and *Hoxa11* for several reasons. First, among Hox mRNAs, the translation mechanism used for each is distinct (*Xue et al., 2015*), *Hoxa11* mRNA translation being RpL38-dependent while *Hoxa3* is RpL38-independent. Second, these two TIEs have very efficient translation inhibition rate and third, their relatively short lengths were more suitable for our experiments. First, we determined their secondary structure by chemical probing assays and then, using cell-free translation extracts and in vivo assays, we deciphered their mode of action. Interestingly, the translation inhibitory mechanism that is mediated by *Hoxa3* TIE is radically distinct from the one used by *Hoxa11*. *Hoxa3* TIE contains an uORF that is translated into a 9 KDa protein through the *Hoxa3* IRES and that requires the presence of the non-canonical translation initiation factor eIF2D. Indeed, eIF2D has been shown to be involved in diverse functions from translation initiation on specific mRNAs to reinitiation and recycling (*Dmitriev et al., 2010*; *Skabkin et al., 2010*; *Weisser et al., 2017*). On the contrary, *Hoxa11* TIE contains a long stable hairpin structure which is preceded by a start-stop codon combination. In mice, these three elements enable a highly efficient inhibition of cap-dependent translation by *Hoxa11* TIE that is achieved through a start-stop stalling mechanism of an 80S ribosome.

## Results

### TIE-mediated inhibition recapitulated in rabbit reticulocyte lysate

To dissect the mechanism of TIE-mediated inhibition, we first set up an efficient cell-free assay. We used rabbit reticulocyte lysate (RRL) and performed in vitro experiments using several reporter constructs. To analyse their translation efficiency, capped mRNAs were translated in RRL in the presence of [$^{35}S$]-methionine to verify the size of the expected synthesized protein and luciferase activity was measured to determine their translation efficiency. In order to standardize the measured translation activities, we inserted the TIEs identified and characterized by *Xue et al., 2015*, in murine *Hoxa3* and mRNAs upstream of the well-characterized human β-globin (*Hbb-b1*) 5'UTR (*Figure 1A*). The *Hbb-b1* allows translation strictly by a cap-dependent mechanism (*Fletcher et al., 1990*). Therefore, in order to evaluate the inhibition due to TIEs, we used a reporter mRNA containing the sole *Hbb-b1* 5'UTR placed upstream of Renilla luciferase (RLuc) coding sequence as a control reporter gene. First, we demonstrated that translation with these extracts is cap-dependent (*Figure 1—figure supplement 1A*). Second, we performed translation experiments with increasing amounts of the same mRNA and determined that with concentrations higher than 100 nM, the translation yield reached a plateau (*Figure 1—figure supplement 1A*). In order to avoid any titration effect due to excess of mRNA, we performed all translation assays at an mRNA concentration of 50 nM, which enables sub-saturating conditions. *Hoxa3* and *Hoxa11* TIEs efficiently promote a translation inhibition by 79% and 91%, respectively (*Figure 1A*). Therefore, our cell-free translation assay using RRL efficiently recapitulates the previously described TIE-mediated translation inhibition (*Xue et al., 2015*). Likewise, TIE-mediated inhibition was efficiently recapitulated in other cell-free translation systems, which are partially or fully cap-dependent from other organisms like wheat germ extracts, *Drosophila* S2 cell extracts, and human HeLa cell extracts (*Figure 1—figure supplement 1B*). Moreover, the same constructs were introduced in a plasmid that allowed us to monitor translation inhibition in vivo. We tested two cell lines that have been shown to express Hox genes, namely human HEK293T

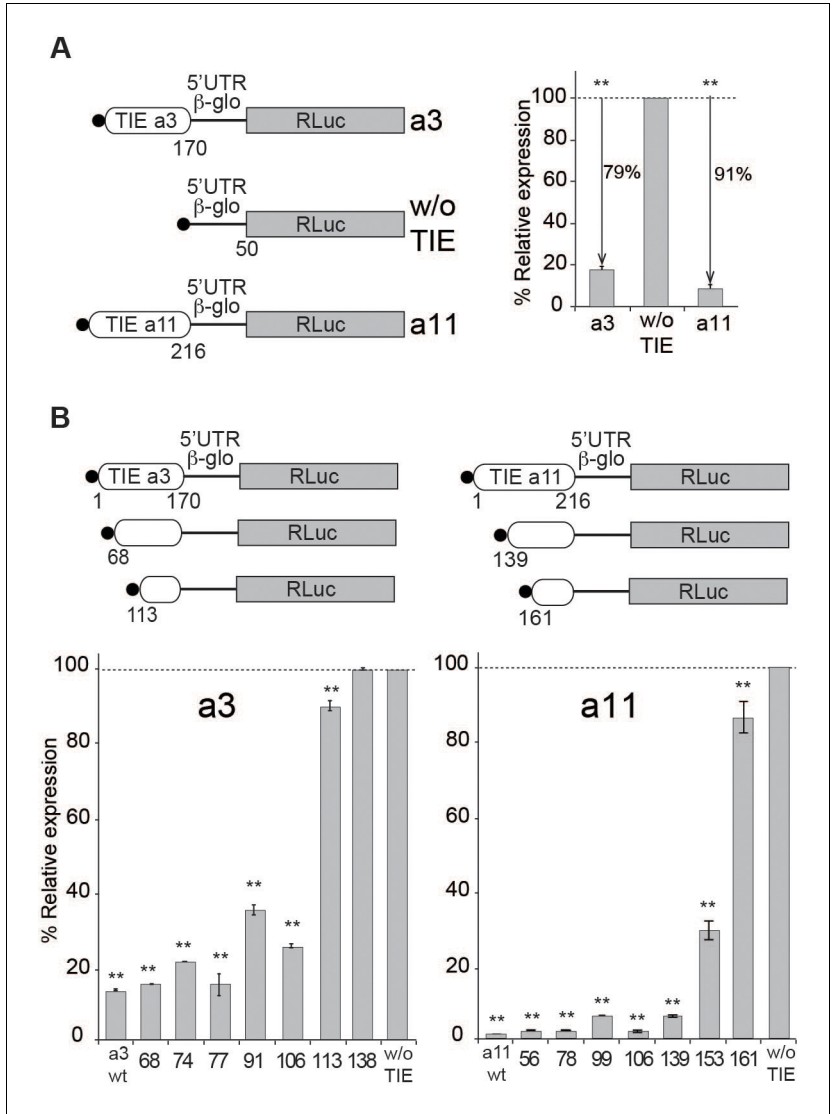

**Figure 1.** Translation inhibitory element (TIE)-mediated inhibition is recapitulated in rabbit reticulocyte lysate (RRL) and does not require full-length TIEs. (A) Three capped mRNAs were used to test TIE-mediated inhibition in vitro. Hoxa3 TIE and *Hoxa11* TIE were placed upstream of the 5'UTR of *Hbb-b1 and the Renilla luciferase coding sequence. Translation assays were performed* in vitro using RRL at an mRNA concentration of 50 nM, which enables sub-saturating conditions. The relative expression of luciferase protein reflects the efficiency of translation inhibition by *Hoxa3* TIE and *Hoxa11* TIE. Values were normalized to that of the control (w/o TIE) which corresponds to normal expression without inhibition and was set to 100%. \*\*p<0.01 (t-test as compared to w/o TIE). n = 3. Experiments were performed in triplicates. (B) Sequential deletions in the 5' extremity of *Hoxa3* TIE and *Hoxa11* TIE constructs were performed to assay their effect on translation. Values of translation expression were normalized to that of the control (w/o TIE). Experiments were performed in triplicates. The percentages of inhibition for each TIE are indicated in the histogram. \*\*p<0.01 (t-test as compared to w/o TIE). n = 3. Experiments were performed in triplicates.

The online version of this article includes the following figure supplement(s) for figure 1:

**Figure supplement 1.** Translation inhibitory element (TIE)-mediated inhibition recapitulated in different in vitro and in vivo systems.

and murine mesenchymal stem cell line C3H10T1/2 (*Phinney et al., 2005*). In these cells, translation inhibition by *Hoxa3* and *Hoxa11* TIE is 98% and 88%, respectively (*Figure 1—figure supplement 1C*). Next, to define the minimal active domains of *Hoxa3* and *Hoxa11*, sequential 5' deletions were performed. In each experiment, translation for each construct was compared to the control (w/o TIE)

(*Figure 1B*). First, we tested large deletions in order to roughly determine the functional domain (*Figure 1—figure supplement 1D*), then more precise deletions to map exactly the 5' end of functional domains for *Hoxa3* and *Hoxa11* (*Figure 1B*). Deletion of 68 nts in *Hoxa3* TIE does not affect inhibition (81%). A further deletion of 113 nts completely abolishes the translation inhibition of *Hoxa3* TIE (10%). Therefore, the minimal *Hoxa3* element encompasses nucleotides 68–170. Concerning *Hoxa11* TIE, the situation is not as clear. In fact, the inhibition is completely lost when 161 nts are deleted. In this case, the deletions indicate that elements essential for translation inhibition are most likely located between residues 139 and 161. In conclusion, these experiments allowed the localization of essential RNA elements required for *Hoxa3* and *Hoxa11* RNA regulons that are required to retain their full inhibitory function.

## TIEs have distinct secondary structures

To gain further insights into the structural and functional properties of TIEs, we performed chemical probing using dimethyl sulfate (DMS) and 1-cyclohexyl-3-(2-morpholinoethyl) carbodiimide metho-*p*-toluene sulfonate (CMCT) reagents for both *Hoxa3* TIE and *Hoxa11* TIE (*Figure 2*). Since modifications were performed in triplicates, the average of reactivity was calculated for each nucleotide at a specific position (*Figure 2—figure supplement 1*). With this reactivity, we built a secondary structural model for both *Hoxa3* and *Hoxa11* TIEs. *Hoxa3* TIE contains a 5' proximal stem loop and another bigger structure comprising two-way junctions (*Figure 2A*). For TIE *a11*, the higher GC content (64%) than *Hoxa3* TIE (45%) suggests more stable structure. Our probing experiments confirmed this statement. It comprises four stem loops and a three-way junction structure (*Figure 2B*). To further validate our models, the secondary structures of both TIEs were also probed in the frame of the full-length Hox 5'UTR which contains the IRES element downstream of the TIE. The reactivities for each nucleotide obtained with isolated TIEs and with TIEs embedded in full-length 5'UTR were highly similar suggesting that the TIEs do fold independently from the IRES (*Figure 2—figure supplement 1*). Using these two models, we wanted to further characterize the structure-function relationships of both TIEs.

## *Hoxa3* TIE contains an uORF that inhibits translation

The 5' truncations experiments allowed us to pinpoint critical elements required for translation inhibition. Interestingly, the minimal *Hoxa3* contains putative uORFs starting from two putative uAUGs at positions 111–113 and 123–125, respectively (*Figure 3A*). In order to test their implication in the translation inhibition, both uAUGs were mutated independently into UAC thereby eliminating any possibility of AUG-like codon recognition. Interestingly, the mutation of AUG111 completely abolishes the inhibition by *Hoxa3* TIE thereby confirming its implication. On the contrary, this is not the case for AUG123 (*Figure 3A*). This is in good agreement with the fact that AUG111 has an optimal Kozak sequence (A at −3 and G at +4), thereby the ribosome initially recognizes it during scanning along the 5'UTR *Kozak, 1986*. To further confirm the assembly of the ribosome on uAUG111, we mutated it into AUG-like codons. We tested previously described AUG-like codons such as AUU, CUG, GUG, and ACG. None of the tested AUG-likes are used for translation inhibition indicating that *Hoxa3* TIE requires a genuine AUG start codon (*Figure 3—figure supplement 1A*, left panel). Since the AUG111 is involved in a stem, it is possible that the absence of inhibition with the AUG-likes is actually due to the disruption of the stem. To rule out this possibility, we mutated the AUG111 into CUG and GUG and inserted simultaneously the compensatory mutations in AUG123 into AGG and ACG respectively that enable the formation of the stem with the mutated CUG and GUG codons (*Figure 3—figure supplement 1A*, right panel). In both mutants, the inhibition was fully destroyed indicating that efficient inhibition requires a genuine AUG start codon independently of the secondary structure (*Figure 3—figure supplement 1A*). Next, we confirmed the use of uAUG111 by toe printing assay. As expected, a canonical +16 reverse transcription (RT) arrest from uAUG111 was clearly detected with GMP-PNP, a non-hydrolysable analogue of GTP that prevents subunit joining (*Eliseev et al., 2018*), and less efficiently with cycloheximide, a translocation inhibitor that binds the E-site of the ribosome (*Garreau de Loubresse et al., 2014*; *Schneider-Poetsch et al., 2010*). Accordingly, when the uAUG is mutated to UAC, the +16 toe print disappears (*Figure 3B*). Altogether, this data confirm that a pre-initiation complex efficiently assembles on the uAUG111. We then wondered whether the ribosome assembled on this AUG codon proceeds to translation

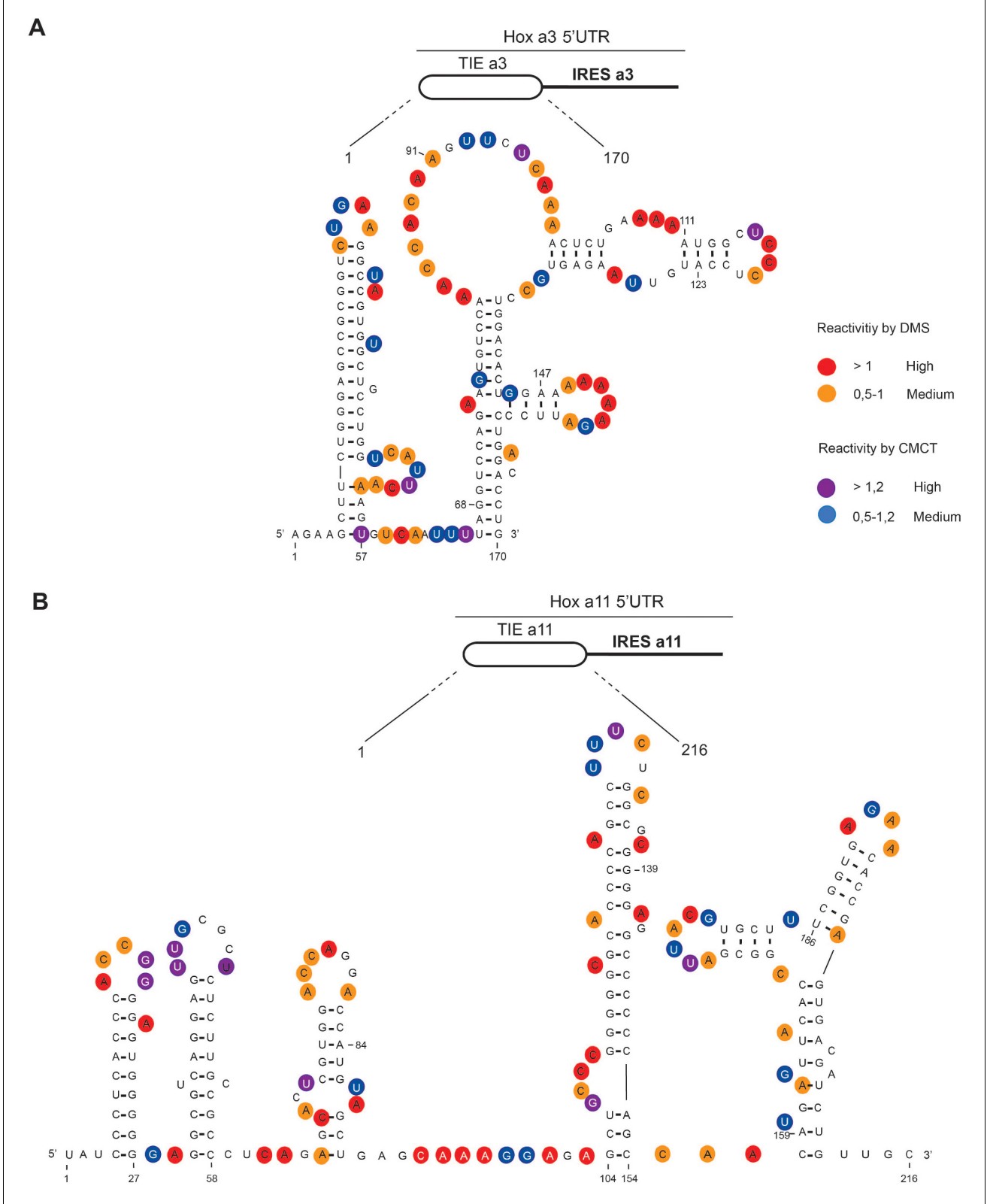

**Figure 2.** The secondary structural models of *Hoxa3* translation inhibitory element (TIE) and *Hoxa11* TIE reveal distinct structures. The structures of (A) *Hoxa3* TIE (170 nucleotides) and (B) *Hoxa11* TIE (216 nucleotides) were obtained by chemical probing using base-specific reagents, dimethyl sulfate (DMS) and 1-cyclohexyl-3-(2-morpholinoethyl) carbodiimide metho-*p*-toluene sulfonate (CMCT). After modifications, reverse transcription was performed using fluorescently labelled primers to determine the position of modified nucleotides. Experiments were performed in triplicates.

*Figure 2 continued on next page*

*Figure 2 continued*

Reactivities are shown as average reactivity from three independent experiments. A representation of reactivities is assigned as colour code depending on a range of values as shown in the figure legend on the right. Reactivity values for each nucleotide with corresponding standard deviations are shown in figure supplements (S2A, S2B, S2C, and S2D).

The online version of this article includes the following figure supplement(s) for figure 2:

**Figure supplement 1.** Average of reactivities of dimethyl sulfate (DMS) and 1-cyclohexyl-3-(2-morpholinoethyl) carbodiimide metho-*p*-toluene sulfonate (CMCT) for *Hoxa3* translation inhibitory element (TIE) and *Hoxa11* TIE.

elongation resulting in a polypeptide synthesis encoded by the uORF. By performing sucrose gradient analysis, we detected polysome formation suggesting that efficient translation is occurring from uAUG111 (*Figure 3—figure supplement 1B*). Accordingly, mutation of uAUG111 drastically reduces the amount of polysomes and the formation of 48S complexes in the presence of GMP-PNP thereby corroborating that the translation is starting on uAUG111 (*Figure 3—figure supplement 1B*). In *Hoxa3* 5'UTR, the uORF starting from uAUG111 is extending through the full IRES. To check further whether the uORF is indeed translated through the full-length *Hoxa3* 5'UTR, we first deleted a single nucleotide (G333) to change the frame and thereby producing a fusion protein formed by the peptide produced from the uORF and RLuc. Indeed, with this single frameshifting point mutation, we could detect a longer protein demonstrating that the pre-initiation complex assembled on the uAUG111 indeed proceeds to translation elongation and is efficiently translating through the full-length 5'UTR of *Hoxa3* mRNA (*Figure 4A*). We also verified the translation from uAUG111 with our reporter constructs containing only the *Hoxa3* TIE. Likewise, the insertion of a single frameshifting nucleotide (A220) allows the detection of a longer fusion protein (*Figure 4—figure supplement 1A*). Remarkably, with a double mutant that combines the mutation of the uAUG111 to ACG with the (A220) insertion, a mixture of two proteins is detected, the fusion protein and the more abundant RLuc protein. Indeed, this confirms that the ribosome requires the uAUG111 but still able to recognize, although less efficiently, the AUG-like ACG as a start codon (*Figure 4—figure supplement 1A*). Similarly, when we use the native full-length *Hoxa3* mRNA, we detect the *Hoxa3* uORF protein of 9 KDa size (*Figure 4—figure supplement 1B*). Altogether, our cell-free translation assays demonstrate that *Hoxa3* TIE translation is achieved by translation of an uORF that starts at AUG111 codon that extends through the whole *Hoxa3* 5'UTR. To confirm these results in vivo, we generated reporter constructs in the plasmid pmirGlo that contains TIEs upstream of Renilla luciferase (hRLuc). Values were normalized to that of control enhanced Firefly luciferase (luc2) to calculate the luciferase activity for each report (*Figure 4B*). As expected, wild-type *Hoxa3* TIE blocks translation very efficiently in both HEK293T and C3H10T1/2 cells. When the uAUG111 is mutated to UAC, the inhibition is significantly affected in both HEK293T and C3H10T1/2 cells at respectively 57% and 45% compared to wild-type *Hoxa3* TIE (*Figure 4B*). Although the inhibition is not fully abolished, these experiments confirmed that the codon AUG111 is critical for efficient translation inhibition in *Hoxa3* in vivo.

### *Hoxa11* TIE-mediated inhibition is mediated by a stalled 80S ribosome

We next asked whether *Hoxa11* TIE has a similar inhibitory mechanism. Deletion experiments suggested that critical elements were located in the region 139–216. We also found that *Hoxa11* TIE contains two putative upstream AUGs at positions 84–86 and 159–161. Mutations of both AUG84 and AUG159 had no impact on translation inhibition (*Figure 5A*). According to our 2D model, a long GC-rich stable stem loop (SL) structure ($\Delta G = -25.00$ kcal/mol) spans nucleotides 104–154 (*Figure 2B*). This long hairpin comprises 16 G-C base pairs that can putatively interfere with the progression of a pre-initiation scanning complex. To test the inhibitory efficiency of this stem loop, we transplanted it in a strictly cap-dependent reporter mRNA containing the 5'UTR of *Hbb-b1* upstream of the luciferase coding sequence. The *Hoxa11* SL was inserted in the middle of the 5'UTR. In this construct the first 25 nucleotides from the 5' proximity are unfolded thereby ensuring proper access to the 5' cap. Interestingly, the translation of this mRNA was significantly abolished showing that this SL on its own is sufficient to inhibit cap-dependent translation when transplanted in another mRNA (*Figure 5—figure supplement 1*). Strikingly, in *Hoxa11* TIE, uAUG84 codon is located 19 nts upstream of the inhibiting SL. This distance is compatible with the assembly of a pre-initiation

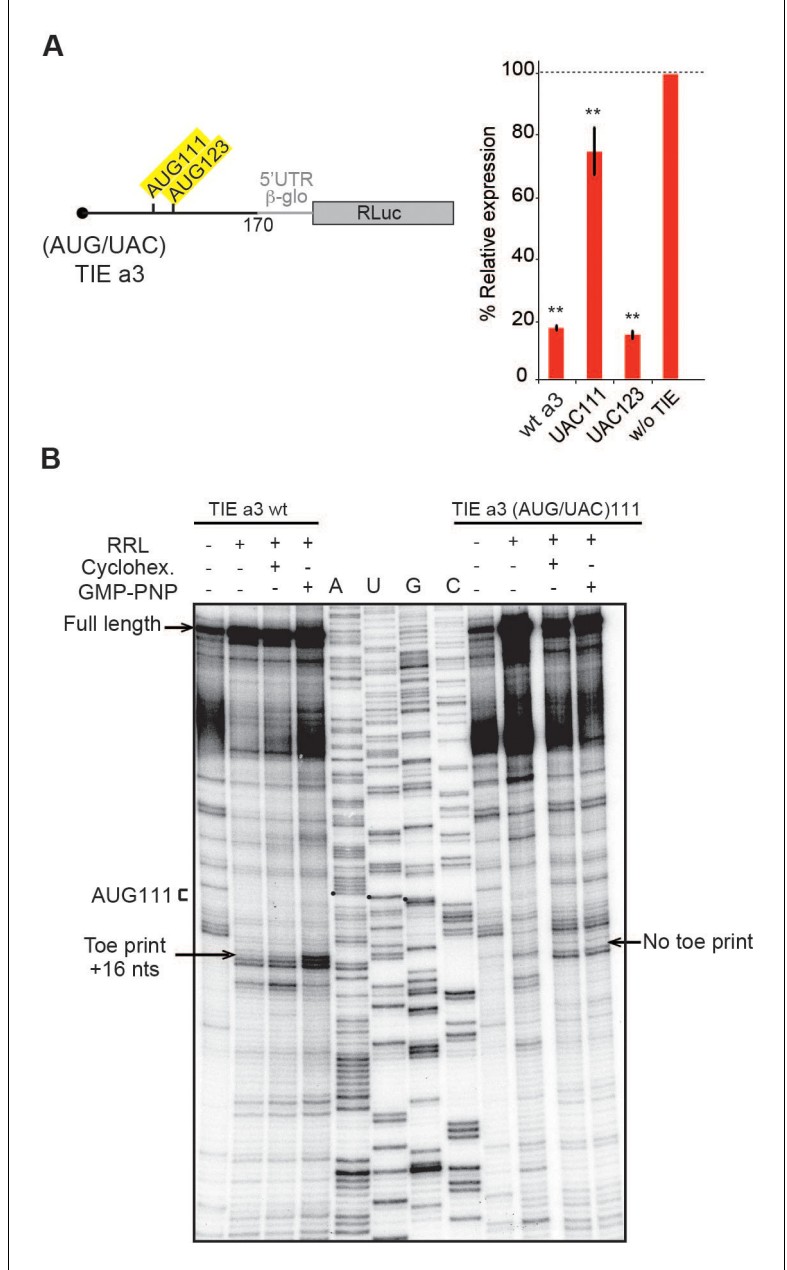

**Figure 3.** Upstream AUG111 in *Hoxa3* translation inhibitory element (TIE) is essential for inhibition. (**A**) Substitution mutations in uAUG111and uAUG123 to UAC in *Hoxa3* TIE were performed. Constructs with the corresponding mutations were translated in rabbit reticulocyte lysate (RRL) and luciferase assay was performed to evaluate the effect of mutation on translation efficiency as previously described. **p<0.01 (t-test as compared to construct w/o TIE). n = 3. Experiments were performed in triplicates. (**B**) Toe printing analysis of ribosomal assembly on two mRNAs, *Hoxa3* TIE Wt and the mutant of upstream (AUG/UAC)111. Initiation complexes were assembled in RRL extracts in the absence or presence of translation inhibitors: cycloheximide and GMP-PNP. Reaction samples were separated on 8% denaturing PAGE together with the appropriate sequencing ladder. Toe print positions were counted starting on the A + 1 of the AUG codon at +16 position. A, U, and G nucleotides of the start codon are marked by black dots. Full-length cDNAs are indicated by an arrow at the top of the gel.

The online version of this article includes the following figure supplement(s) for figure 3:

**Figure supplement 1.** Upstream AUG111 in *Hoxa3* translation inhibitory element (TIE) is essential for inhibition.

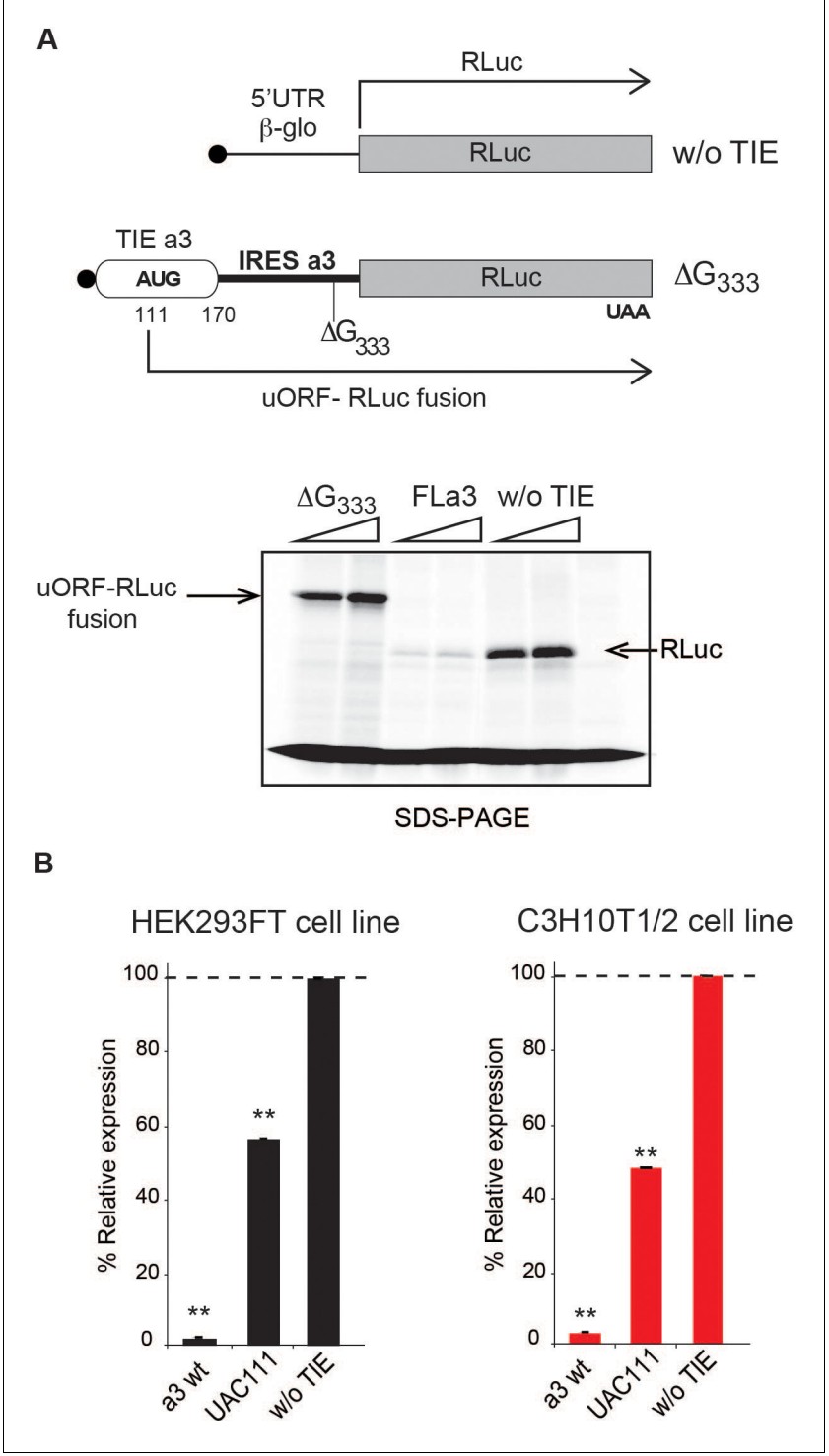

**Figure 4.** The uAUG111 in *Hoxa3* translation inhibitory element (TIE) is translated through 5'UTR of *Hoxa3*. (A) Three transcripts were used for this experiment: full-length 5'UTR of *Hoxa3*, a deletion mutant at nucleotide $G_{333}$ in *Hoxa3* internal ribosome entry site (IRES) and a control transcript without TIE. To test the translation of upstream open reading frame (uORF) in *Hoxa3* TIE starting form uAUG111, a deletion of G in *Hoxa3* IRES at position 333 was performed to create a longer uORF that is in the same frame as the ORF of Renilla luciferase to create an N-terminally extended luciferase. FL*a3* mRNA was used as a control to correspond to the full-length *Hoxa3* 5'UTR (TIE + IRES). Transcripts were translated in vitro in rabbit reticulocyte lysate (RRL) and products were loaded on 10% SDS-PAGE in the presence of $^{35}S$-methionine. (B) In vivo luciferase assays in two embryonic cells

*Figure 4 continued on next page*

*Figure 4 continued*

lines: HEK293FT (left) and C3H10T1/2 (right). Reporter constructs in pmirGlo containing *Hoxa3* TIE, u(AUG/UAC) 111, and without TIE were transfected in the two indicated cell lines. Renilla luciferase expression was normalized to the control (w/o TIE), which was set to 100%. **p<0.01 (t-test as compared to empty plasmid [w/o TIE]). Experiments were performed in triplicates. n = 3.

The online version of this article includes the following figure supplement(s) for figure 4:

**Figure supplement 1.** Translation of *Hoxa3* translation inhibitory element (TIE) upstream open reading frame (uORF) in different constructs.

---

complex on uAUG84 without clashing with the SL. Moreover, the distance between the uAUG and the SL is optimal for favouring AUG recognition by scanning arrest of the pre-initiation complex forced by the SL. Importantly, the uAUG84 is immediately followed by a stop codon UAG87. The sequence context of the uAUG84 (A at −3 and U at +4) is suboptimal compared to the consensus Kozak sequence. This unique combination of start-stop codon upstream of SL structure raised the question of whether the ribosome is forced to recognize this uAUG despite a suboptimal Kozak context. To address this hypothesis, we mutated the stop codon UAG87 to UGG thereby creating an uORF. When the stop is mutated, a small peptide is produced from the translation of uAUG84 through 5'UTR, which we called *Hoxa11* uORF (*Figure 5B*). This experiment demonstrates that uAUG84 is indeed efficiently used as a start codon despite its suboptimal sequence context. Unfortunately, the presence of the highly stable SL is not compatible with the toe printing assay. Indeed, premature RT arrests occur due to the presence of the highly stable SL, rendering the toe printing assay on TIE *Hoxa11* impossible. Therefore, to further confirm that the ribosome is efficiently assembled on uAUG84, we performed sucrose gradient analysis with radiolabelled mRNAs. With wild-type *Hoxa11* TIE, an 80S complex efficiently accumulates in the presence of cycloheximide as expected. However, an 80S complex is also detected without inhibitor indicating that the 80S complex is in fact a stalled ribosome (*Figure 5C*, *Figure 5—figure supplement 2*). The mutation of uAUG84 to UAC drastically reduces the amount of 80S complex; in contrast, mutation of uAUG159 to UAC does not affect 80S accumulation. This further confirms that the stalled 80S complex is indeed assembled on the uAUG84. Altogether, this data show that a stalled ribosome is indeed assembled on uAUG84 and the stalling is caused by the synergistic effect of a stop codon next to the AUG and a stable SL downstream of the start-stop of *Hoxa11*.

## Mass spectrometry analysis pre-initiation complexes programmed by *Hoxa3* and *Hoxa11* TIE

To further characterize the two different modes of action employed by *Hoxa3* and *Hoxa11*, we identified the factors specifically acting in such mechanisms. For that, we purified pre-initiation complexes programmed by *Hoxa3* and *Hoxa11* TIE suitable for mass spectrometry analysis. Briefly, ribosomes were assembled on chimeric biotinylated mRNA–DNA molecules and immobilized on streptavidin-coated beads after incubation with RRL in the presence of cycloheximide or GMP-PNP. Complexes were then eluted by DNase treatment as previously described (*Chicher et al., 2015*; *Prongidi-Fix et al., 2013*) and analysed by mass spectrometry. Three different mRNA constructs were used, *Hoxa3* TIE upstream of 5'UTR *Hbb-b1*, *Hoxa11* TIE upstream of 5'UTR *Hbb-b1*, and 5'UTR *Hbb-b1* as a negative control. Comparison between the three mRNAs blocked with cycloheximide enabled the identification of specific factors interacting with each RNA (*Figure 6*). Interestingly, for *Hoxa3* TIE, we identified a set of translation-related proteins. Among the strongest hits, we found eIF2D, a non-canonical GTP-independent initiation factor which has been shown to be involved in the initiation step on specific mRNAs (*Dmitriev et al., 2010*; *Vaidya et al., 2017*; *Vasudevan et al., 2020*), reinitiation on main ORF (*Ahmed et al., 2018*; *Weisser et al., 2017*), and recycling after translation termination (*Skabkin et al., 2013*; *Skabkin et al., 2010*; *Young et al., 2018*). Another interesting hit is methionine aminopeptidase MetAP1 which removes N-terminal methionine from nascent proteins in a co-translational manner (*Varland et al., 2015*).

Other factors that are linked to translation have been selected such as the scanning factor eIF1A and its isoform eIF1A-X, eIF3a, arginyl- and leucyl-tRNA synthetases QARS and LARS, DEAD-box helicases DHX36 and DDX39B, elongation factor HBS1L, and RpL38 ribosomal protein. For *Hoxa11* TIE, we identified different translation-related proteins among which are ASAP1, a GTPase activator

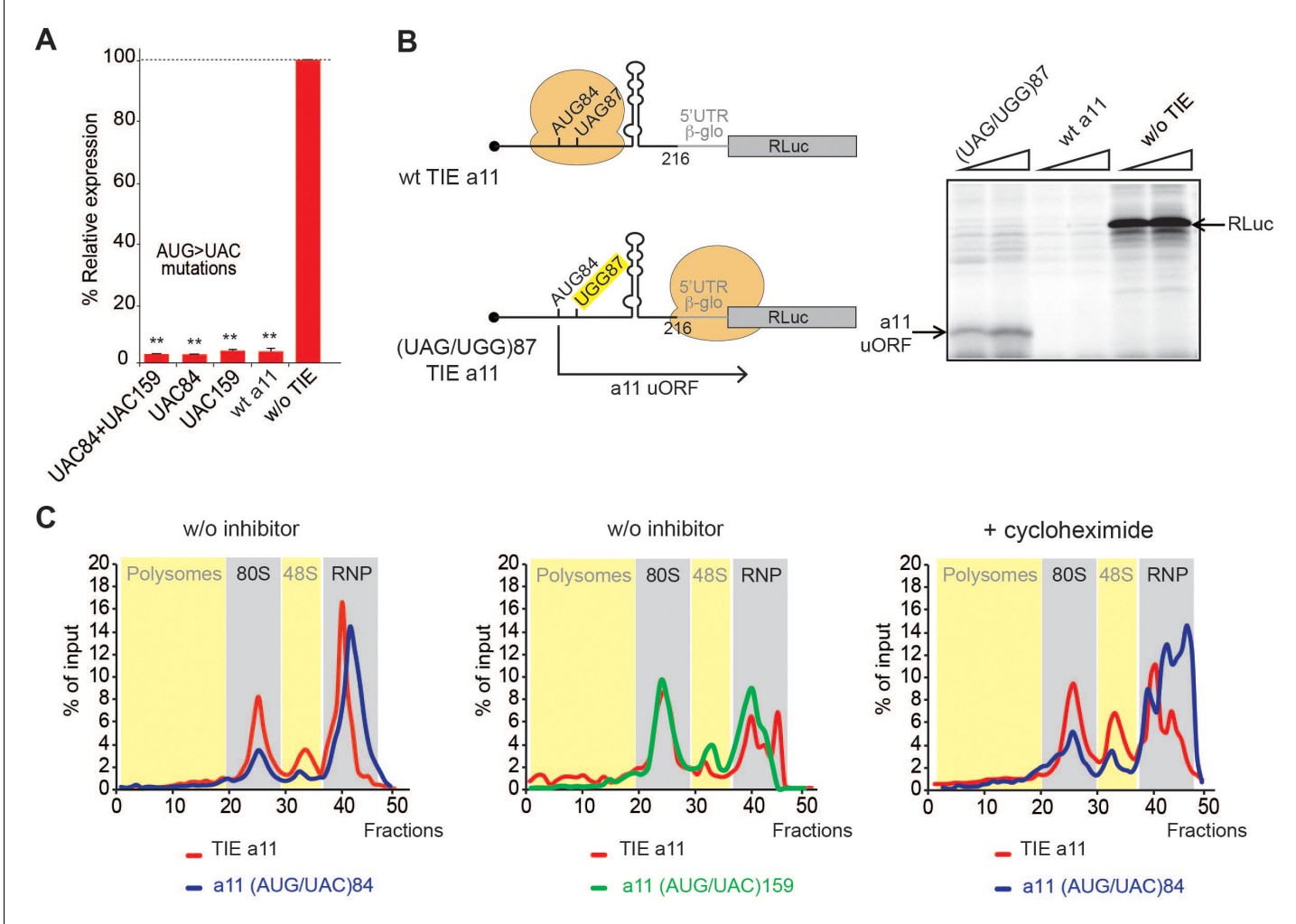

**Figure 5.** A start-stop upstream open reading frame (uORF) in *Hoxa11* translation inhibitory element (TIE) stalls an 80S upstream of a highly stable structure. (**A**) Mutational analysis of uAUGs in *Hoxa11* TIE. Three transcripts with AUG/UAC mutations were used: M1: (AUG/UAC)84 + (AUG/UAC)159, M2: (AUG/UAC)84, and M3: AUG/UAC159. Transcripts were translated in rabbit reticulocyte lysate (RRL) at 50 nM concentrations and the luciferase expression was normalized to the control (w/o TIE) as previously described. **p<0.01 (t-test as compared to construct w/o TIE). n = 3. Experiments were performed in triplicates. (**B**) A single substitution (A/G)88 mutation destroys the stop codon UAG87 to UGG in *Hoxa11* TIE, *Hoxa11* TIE wt, and control (w/o TIE) were also translated as references in RRL. Translation products were loaded on 15% SDS-PAGE. (**C**) Ribosomal pre-initiation complexes were assembled and analysed on 7–47% sucrose gradient with [alpha-$^{32}$P]GTP-radiolabelled *Hoxa11* TIE as well as the two mutants of uAUG/UAC at the previously indicated positions in the absence or presence of cycloheximide. Heavy fractions correspond to polysomes and lighter fractions correspond to free RNPs. The coloured sections corresponding to RNP, 48S, 80S, and polysomes have been assigned according the the OD$_{254nm}$ profiles (see also Figure supplement S6).

The online version of this article includes the following figure supplement(s) for figure 5:

**Figure supplement 1.** Transplanting *Hoxa11* translation inhibitory element (TIE) stem loop structure in *Hbb-b1* 5' UTR efficiently inhibits translation of Renilla luciferase (RLuc) mRNA.

**Figure supplement 2.** Polysome fractionation on 7–47% sucrose gradient of pre-initiation complexes.

protein, MetAP1, RpL38, Valyl-tRNA synthetase, and eIF3j, another subunit of initiation factor eIF3 usually dissociating at early stages of initiation to allow mRNA entry (*Aylett et al., 2015*; *Fraser et al., 2007*; *Young and Guydosh, 2019*), thereby unlikely to be present in initiation complexes. When comparing *Hoxa3* TIE with *Hoxa11* TIE, we could detect some initiation factors and translation-related proteins specific for *Hoxa3* like eIF2D, eIF1A, eIF1A-X, eIF5B, LARS, DHX21, and HBS1L (*Figure 6*, *Supplementary file 1*). Interestingly, HBS1L is enriched in both *Hoxa3* and *Hoxa11* cycloheximide-blocked complexes. HBS1L is a member of translational GTPase family which

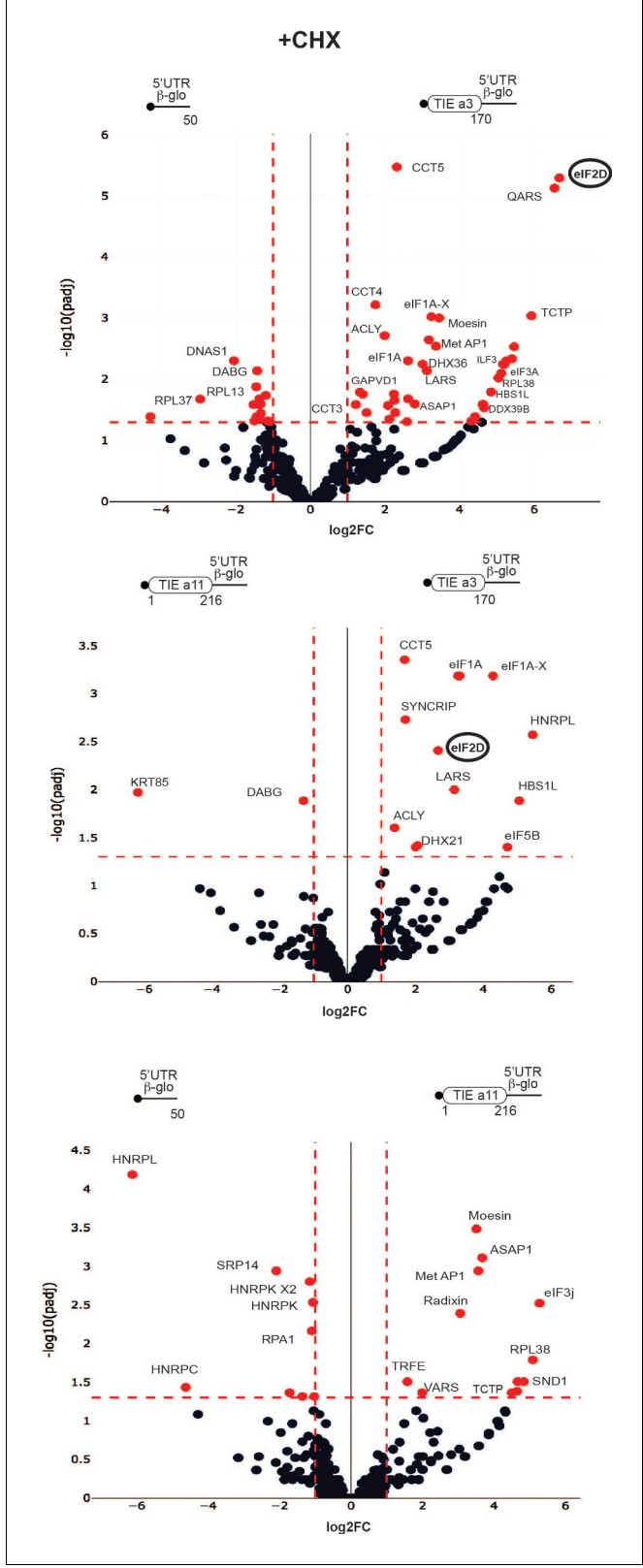

**Figure 6.** Distinct profiles for factors involved in translation inhibitory element (TIE)-mediated inhibition blocked with cycloheximide. Mass spectrometry analysis of cycloheximide-blocked translation initiation complexes and on three transcripts: *Hoxa3* TIE, *Hoxa11* TIE placed upstream of 5'UTR of *Hbb-b1*, and the 5'UTR of *Hbb-b1* (control). Graphical representation of proteomics data: protein log$_2$ spectral count fold changes (on the x-axis) and the

*Figure 6 continued on next page*

*Figure 6 continued*

corresponding adjusted $\log_{10}$p-values (on the y-axis) are plotted in a pair-wise volcano plot. The significance thresholds are represented by a horizontal dashed line (p-value=1.25, negative binomial test with Benjamini–Hochberg adjustment) and two vertical dashed lines (−1.0-fold on the left and +1.0-fold on the right). Data points in the upper left and upper right quadrants indicate significant negative and positive changes in protein abundance. Protein names are labelled next to the off-centred spots and they are depicted according to the following colour code: red spots are significant hits and black are non-significant with <10 spectra. Data points are plotted based on the average spectral counts from triplicate analysis. Three profiles were produced by comparing the proteomics of two transcripts.

The online version of this article includes the following figure supplement(s) for figure 6:

**Figure supplement 1.** Distinct profiles for factors involved in translation inhibitory element (TIE)-mediated inhibition blocked with GMP-PNP.

---

transports Pelota to stalled ribosomes with an empty A-site or an mRNA-occupied A-site in a codon-independent manner (*Lee et al., 2007*; *Shao et al., 2016*; *Shoemaker et al., 2010*). Another interesting hit is eIF5B, a factor whichcatalyses the joining of the large ribosomal subunit independently of its GTPase activity (*Wang et al., 2019*). The GTPase activity is only required for eIF5B release from the assembled 80S and thereby allows to proceed to elongation. EIF5B assists the correct positioning of Met-tRNA$^{Met}$ in the P-site (*Wang et al., 2020*). Interestingly, RpL38 protein is enriched in both TIE-programmed complexes. In fact, all the ribosomal proteins are present in the purified complexes (*Figure 6*, *Supplementary file 1*) so this enrichment might be due to free RpL38 outside of the ribosome particle. This is particularly interesting because RpL38 is involved for Hox IRES elements (*Xue et al., 2015*). Altogether, these results show a variation in the translation factors involved which hints two distinct TIE-mediated inhibitions.

Similarly, we purified programmed pre-initiation complexes blocked by GMP-PNP (*Figure 6—figure supplement 1*, *Supplementary file 2*). By comparing *Hoxa3* to either *Hbb-b1* or *Hoxa11* mRNAs, we also found an enrichment of eIF2D as *Hoxa3*-specific factor. Interestingly, we also found an enrichment of PKR protein (also called eIF2AK2) for both *Hoxa3* and *Hoxa11* (*Figure 6—figure supplement 1*). PKR is known to bind double-stranded RNA during viral infection which mediates its auto-activation and induces the phosphorylation of eIF2α subunit (*Adomavicius et al., 2019*). This leads to the inhibition of mRNA translation. The enrichment of both eIF2D and PKR for *Hoxa3* raised the question of whether eIF2D is indeed mediating the translation of uORF by an alternative mechanism. To address this hypothesis and the involvement of eIF2D, we conducted a set of in vivo experiments.

## Translation inhibition by TIE requires eIF2D

After specifically identifying eIF2D by mass spectrometry analysis for *Hoxa3* TIE-ribosomal complex, we were interested in getting more insights into how this factor might be involved. Previous studies have shown that eIF2D is a non-canonical translation initiation factor which delivers tRNA to the P-site of the ribosome in a GTP-independent manner (*Dmitriev et al., 2010*). It has been shown to be involved in the initiation step on specific mRNAs (*Dmitriev et al., 2010*; *Vaidya et al., 2017*; *Vasudevan et al., 2020*), reinitiation on main ORF (*Ahmed et al., 2018*; *Weisser et al., 2017*). Additionally, other studies showed that eIF2D is required for recycling after translation termination (*Skabkin et al., 2013*; *Skabkin et al., 2010*; *Young et al., 2018*). To confirm the involvement of eIF2D in *Hoxa3* TIE-mediated inhibition mechanism, we performed a co-transfection assay in HEK293T cell line using siRNA against either eIF2D or non-target pool of siRNAs (siRNA-NT) together with a reporter plasmid harbouring *Hoxa3* TIE. The plasmid used harbours two reporter genes, RLuc to test the impact of TIEs and Firefly luciferase for normalization purposes (*Figure 7A*). As a control, we used reporter plasmids with *Hoxa11* TIE and the same plasmid without TIE. After 48 hr incubation, cells were lysed for western blot analysis and luciferase assay (*Figure 7A*). For normalization, RLuc values were compared to that of Firefly luciferase (RL/FL) in each construct. To determine the effect of eIF2D silencing, the RL/FL ratio was compared to that with the siRNA-NT, which serves as a control for siRNA effect on the cell line. Thereby, the comparison between the two values would determine the effect of eIF2D silencing with each TIE. Interestingly, silencing eIF2D in *Hoxa3*-transfected cells at an efficiency of 76% drastically abolishes *Hoxa3*-mediated inhibition and

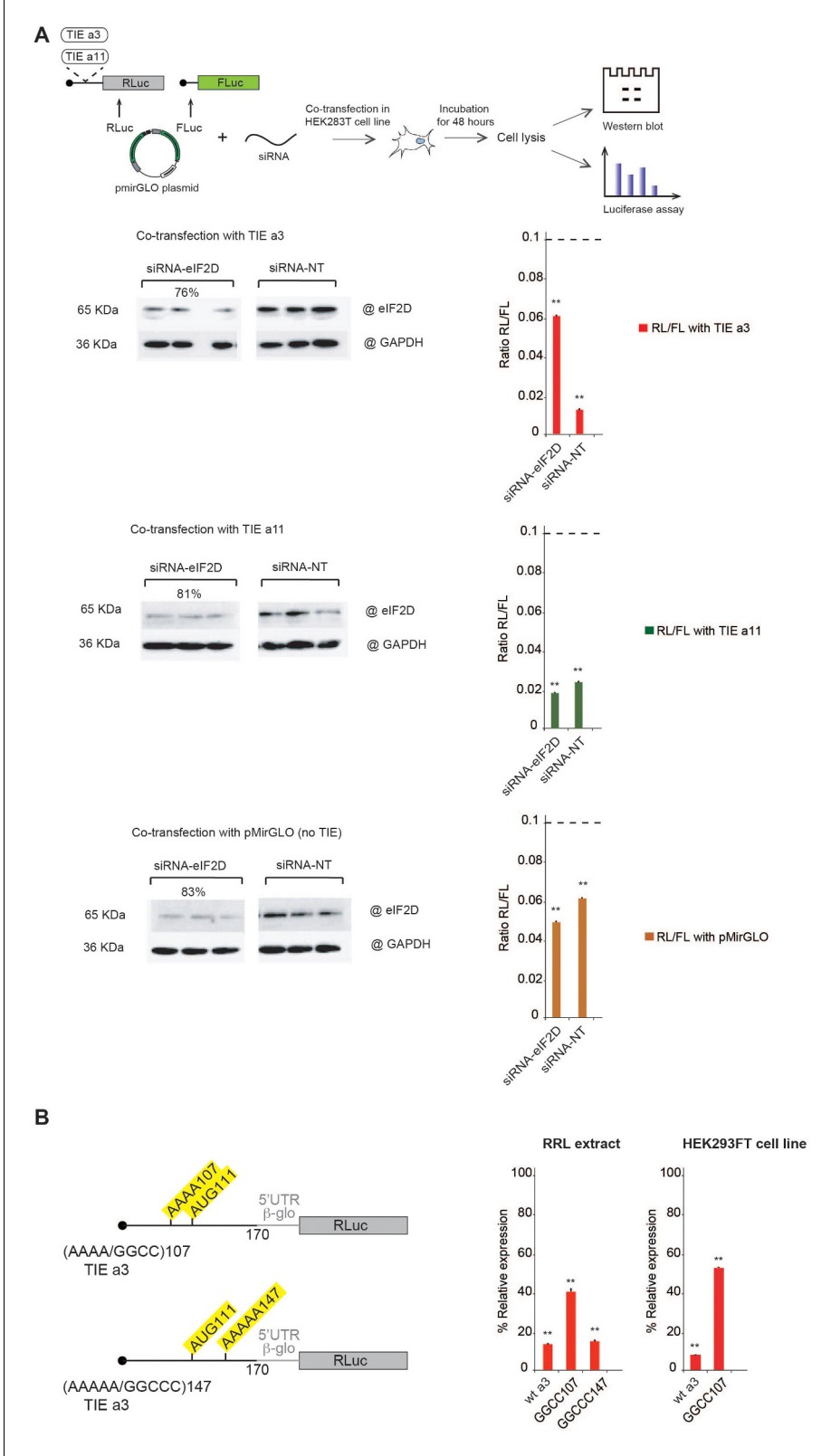

**Figure 7.** Co-transfection assay of translation inhibitory element (TIE) plasmids and siRNA against eIF2D confirms its implication in *Hoxa3*-mediated inhibition. Mutational analysis of A-motif sequences in *Hoxa3* TIE shows a requirement for an upstream A-motif for efficient inhibition. (A) Co-transfection assay was performed using pmirGLO plasmids with inserts of *Hoxa3* TIE, *Hoxa11* TIE, and w/o TIE (pmirGLO) with siRNA against eIF2D (siRNA-eIF2D), and non-target pool of siRNA (siRNA-NT) in HEK293 cell line. The plasmids harbour two reporter genes, Renilla luciferase for monitoring

*Figure 7 continued on next page*

*Figure 7 continued*
the impact of TIEs and Firefly luciferase for normalization. Plates were incubated for 48 hr followed by cell lysis. Cell lysate was analysed by western blot and luciferase assay. The histograms present the Renilla to Firelfly luciferase activity ratio. **p<0.01 (t-test as compared to construct w/o TIE). n = 3. Experiments were performed in triplicates. Efficiency of silencing of each protein was quantified by western blot analysis for *Hoxa3* TIE and *Hoxa11* TIE samples. (B) Two sets of mutations were performed on distinct A-motif sequences in *Hoxa3* TIE. The first mutation is AAAA/GGCC107 and the second mutation AAAAA/GGCCC147. The transcripts were in vitro translated in RRL with *Hoxa3* TIE Wt and control (w/o TIE). Results were confirmed in vivo in HEK293FT cell line. Luciferase activities were normalized to control (w/o TIE). **p<0.01 (t-test as compared to construct w/o TIE). n = 3. Experiments were performed in triplicates.

RL/FL ratio (0.06) increases six times compared to that in siRNA-NT cells (0.01) (*Figure 7A*). On the contrary, the silencing of eIF2D at efficiency of 81% has no significant effect on *Hoxa11*-mediated inhibition (RL/FL ratio of 0.02 with both siRNAs). Similarly, silencing eIF2D at 83% efficiency has no effect on general translation (RL/FL ratio remains around 0.06 with both siRNAs). Altogether, this confirms the specific requirement of eIF2D for efficient *Hoxa3* inhibition but not for *Hoxa11*. This conclusion is in good agreement with our previous mass spectrometry analysis of *Hoxa3*-ribosomal complexes blocked by translation inhibitors, which show the specific presence of eIF2D in the initiation complexes programmed with *Hoxa3* TIE (*Figure 6*, *Figure 6—figure supplement 1*). The fact that we found eIF2D in pre-initiation complexes suggests that eIF2D is involved in the initiation of *Hoxa3* uORF. Since *Hoxa3* uORF is long and extends downstream the AUG start codon of the main CDS in native *Hoxa3* mRNA, which means that the *Hoxa3* uORF partially overlaps the main CDS, a mechanism using a reinitiation event after *Hoxa3* uORF translation is impossible. Therefore, we rather favour a model in which eIF2D is required for *Hoxa3* initiation. Previous studies have shown that an A-motif upstream of an uAUG has been shown to be important for proper eIF2D recruitment (*Dmitriev et al., 2010*). Interestingly, a closer look at the sequences in the *Hoxa3* TIE uORF revealed A-rich motifs upstream and downstream of uAUG111. We tested the implication of both A-motifs, AAAA107 upstream of the AUG and AAAAA147 downstream of the AUG, in *Hoxa3* TIE-mediated inhibition (*Figure 7B*). In order to avoid any side effect due to mutation in the sequence context around the uAUG, we kept an optimal Kozak sequence and mutated the As at position 107–110 to GGCC thereby keeping a purine residue at −3 position. The second A-motif downstream of AUG was similarly mutated to GGCCC. Interestingly, the mutation of upstream A-motif had a twofold reduction effect on translation inhibition of *Hoxa3* TIE. In contrast, mutation of the downstream A-motif does not affect *Hoxa3*TIE inhibition (*Figure 7B*). Additionally, we confirmed the implication of upstream A-motif in vivo with HEK293T cell line using plasmids with wild-type *Hoxa3* and the mutant of AAAA107 into GGCC (*Figure 7B*). The AAAA107 mutant reduces the inhibitory efficiency of *Hoxa3* TIE by 2.5 times compared to wild-type *Hoxa3*. Therefore, we show that the A-motif upstream of the AUG is critical for translation inhibition and most probably because it is required for eIF2D recruitment.

## Discussion

Our study has shown that two HoxA mRNAs, *Hoxa3* and *Hoxa11*, are regulated by different mechanisms to ensure the inhibition of cap-dependent translation and allowed us to propose two distinct models for their mode of action (*Figure 8*). First, we have shown that TIEs can function in vitro using cell-free translation extracts. We then confirmed the results obtained with these extracts using in vivo assays in several cell lines. Our findings suggest that *Hoxa3* inhibits translation by an uORF which is translated through the whole 5'UTR of *Hoxa3* mRNA producing a small protein of size 9 KDa. Interestingly, the alignment of *Hoxa3* TIE shows a conservation of the uAUG111 (highlighted in red box) among different species. In contrast to the localization of the uAUG that is highly conserved, the coding sequence of the uORF is not conserved among species (*Figure 8—figure supplement 1A*). Indeed, uORFs have been recognized as regulators of translation for number of cellular mRNAs (*Barbosa et al., 2013*). For instance, four uORFs in the 5' leader of yeast *GCN4* mRNA restrict the flow of scanning ribosomes from the cap site to the *GCN4* initiation codon (*Dever et al., 2016*; *Hinnebusch, 1993*). In plants, the uORF in *AdoMetDC* mRNA generates a nascent hexapeptide that interacts with its translating ribosome to suppress translation of *AdoMetDC* RNA in a cell-specific manner (*Uchiyama-Kadokura et al., 2014*). Interestingly, our data indicates that eIF2D is

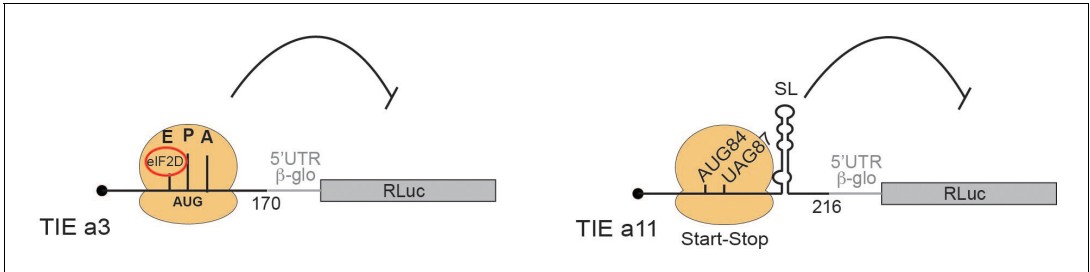

**Figure 8.** Two distinct models for translational inhibition by *Hoxa3* translation inhibitory element (TIE) and *Hoxa11* TIE. A model for *Hoxa3* TIE suggests a ribosomal assembly on the uAUG111 with a requirement of eIF2D initiation factor. The model for *Hoxa11* TIE suggests a stalled 80S ribosome on AUG-stop codons combination upstream of a highly stable structure.

The online version of this article includes the following figure supplement(s) for figure 8:

**Figure supplement 1.** Alignment of *Hoxa3* translation inhibitory element (TIE) and *Hoxa11* TIE among different species shows variation in the conservation of the inhibitory elements.

required in *Hoxa3* TIE mode of action (*Figure 6* and *7A*). A recent study has shown that *Drosophila* ATF4 mRNA translation is induced by eIF2D and its homologue DENR during integrated stress response (*Vasudevan et al., 2020*). In this case, eIF2D requires its RNA binding motif to mediate translation of ATF4 mRNA through its 5' leader sequence consisting of multiple uORFs (*Vasudevan et al., 2020*). Moreover, it has been shown that eIF2α phosphorylation during stages of embryonic development promotes translation from uORFs (*Friend et al., 2015*). Therefore, canonical cap-dependent translation initiation with eIF2 is not possible during embryonic development. The cap-dependent translation initiation of uORF from *Hoxa3* TIE might use eIF2D as an alternative to replace inactive phosphorylated eIF2 to promote uORF translation. So far, the only *cis*-acting sequence that has been clearly defined on an mRNA for specific eIF2D recruitment is an A-rich motif upstream of the start codon (*Dmitriev et al., 2010*). We showed that *Hoxa3* TIE contains such an A-motif upstream of the uORF that is critical for TIE function (*Figure 7*). Other reports showed that eIF2D would form initiation complexes on leaderless and A-rich 5'UTRs (*Akulich et al., 2016*; *Dmitriev et al., 2010*). In *Hoxa3* TIE, mass spectrometry analysis enabled us to demonstrate that eIF2D is present only in pre-initiation complexes programmed with *Hoxa3* TIE. Importantly, two more factors, eIF5B and HBS1L, have been specifically found in *Hoxa3* complexes, which would be interesting to investigate their involvement in eIF2D-dependent mechanism (*Figure 6*). Regarding *Hoxa11* TIE, the ribosome recognizes a combination of two *cis*-acting elements in the 5'UTR (*Figure 8*). (i) A start-stop codon combination located at positions 84–89 and (ii) a long highly stable GC-rich helical structure (SL) located at +20 downstream of the uAUG (by convention, the A from the AUG being +1). These two elements act in synergy to promote the stalling of an 80S complex upstream of the SL. This is a reminiscent of a similar mechanism that has been described in the *Arabidopsis thaliana* NIP5.1 5'UTR mRNA that contains an AUG-stop that regulates translation of the main ORF through a ribosome stalling mechanism and mRNA degradation (*Tanaka et al., 2016*). This mechanism requires only the start-stop codons. In the case of *Hoxa11* TIE, the stable SL structure that is present downstream of the AUG-stop plays a major role. A ribosome that is stalled with a stop codon in the A-site, in other words an empty A-site without any tRNA, is usually the signal for recruitment of the release factors to dissociate the ribosomal subunits from the mRNA. With our cell-free translation assay and sucrose gradient analysis, we showed that the stalled 80S programmed with *Hoxa11* TIE is very stable and does not dissociate (*Figure 5C*). A possible explanation for the stability of this complex could be that the SL blocks the access of the release factors to the empty A-site thereby preventing ribosome dissociation (*Brown et al., 2015*). Interestingly, the alignment of *Hoxa11* TIE among different species shows that the start-stop combination (highlighted in red) lacks conservation and remains species-specific (*Figure 8—figure supplement 1B*). Some species possess a substitution of the start codon AUG to AUG-like codons such as GUG or a mutation of stop codon that leads to a longer uORF. In contrast, the SL (104–154) remains highly conserved suggesting its high functional significance (*Figure 8—figure supplement 1B*). Accordingly, we have shown that the sole SL is strong enough to impede scanning by the pre-initiation complex. Indeed,

secondary structures in the 5'UTR have been shown to inhibit translation like the case of a conserved SL structure in the 5'UTR of TGF-β1 mRNA (*Jenkins et al., 2010*). Therefore, in various species, *Hoxa11* TIE might use different combinations of *cis*-acting elements in order to block cap-dependent translation, the common *cis*-acting element between all species being the SL that is conserved.

As previously described, the *Hoxa11* IRES requires the presence of the ribosomal protein RpL38 while *Hoxa3* IRES is RpL38-independent (*Kondrashov et al., 2011*). Our probing experiments revealed that the folding of both *Hoxa3* and *Hoxa11* TIEs is independent of the presence of IRES suggesting that their mode of action does not depend on the IRES. TIE may have evolved in such a way to favour the translation from the downstream IRES hence justifying why there is variation in terms of sequence and mechanism but same inhibitory effect. This unique combination of an inhibitor of canonical translation mechanism and the activator of specialized translation sets an interesting point on how the 5'UTR elements confer ribosome specificity to translation (*Xue et al., 2015*). Importantly, the acquisition of these TIEs in subsets of Hox mRNAs enables an additional layer of regulatory control between the canonical translation and the IRES-dependent one. One intriguing study would be to determine how other TIEs a4 and a9 inhibit translation and whether there are common functional features amongst all Hox mRNAs. Beyond Hox mRNAs, our data on eIF2D suggests a specific role in translation initiation and it would be interesting to decipher its precise role at the molecular level. More precisely, investigations that will allow determining how uORF length, codon composition, and consensus sequences may influence the role of eIF2D in the initiation process on uORFs are needed. Future studies will be required to fully understand the role of eIF2D in translation initiation of specific mRNAs.

# Materials and methods

## Plasmids

For in vitro studies, murine *Hoxa3* TIE (170 nts) and *Hoxa11* TIE (216 nts) (sequences were kindly provided by Dr Maria Barna, which were amplified from mouse E10.5–12.5 cDNA; *Xue et al., 2015*) were placed upstream of 5'UTR of *Hbb-b1* (accession number: KU350152) (50 nts) and *Renilla reniformis* luciferase coding sequence (accession number: M63501) (936 nts). These constructs were cloned in pUC19 vector in the *Hin*dIII site, then used as a template for further PCR amplifications and site-directed mutagenesis.

For in vivo studies, we introduced an *Eco*RI restriction site upstream of hRLuc-neo fusion sequence in pmirGLO vector (Promega) using Quick Change site-directed mutagenesis kit II XL (Thermo Fischer Scientific). Subsequent cloning experiments of *Hoxa3* TIE, *Hoxa11* TIE, and their mutants were performed in pmirGLO vector at EcoRI site using NEBuilder *HiFi* DNA Assembly kit. All clones were checked by sequencing.

## Cell lines

Two cell lines were used for our in vivo assays: human embryonic kidney cell line HEK293FT (ATCC) purchased from Invitrogen (ref R700-07) and murine mesenchymal stem cell line C3H10T1/2 (clone 8, ATCC CCL26). HEK293FT cells were cultured in Dulbecco's modified eagle medium (DMEM) with 2 mM of L-glutamine and 10% fetal bovine serum (FBS) supplemented with 100 units/ml of penicillin/streptomycin. Subcultures were performed after trypsin-EDTA treatment for dissociation at subconfluent conditions (70–80%) 1:4 to 1:10 seeding at $2–4 \cdot 10^4$ cells/cm$^2$ according to manufacturer's instructions. C3H10T1/2 cells were cultured in basal DMEM supplemented with 2 mM glutamine, 1.5 g/l sodium bicarbonate, and 10% FBS supplemented with 40 µg/ml gentamicine. Subcultures were performed after trypsin-EDTA treatment for dissociation at sub-confluent conditions (60–70%). Seeding dilutions were performed at 2000 cells/cm$^2$ one time per week.

## RNA transcription

Transcription templates were generated by PCR amplification from the plasmids pUC19-TIE. The amplified templates were used for in vitro transcription with recombinant T7 RNA polymerase in the presence of either m$^7$G$_{ppp}$G cap analogue or non-functional cap analogue A$_{ppp}$G (New England Biolabs). To check RNA integrity, an aliquot was mixed with formamide dye and loaded on 4% denaturing polyacrylamide gel. The RNA was visualized under UV light after ethidium bromide staining. To

eliminate unincorporated nucleotides, the RNA sample was loaded on a gel filtration Sephadex G25-column (Pharmacia Fine Chemicals), proteins were then eliminated by phenol extraction, and the RNA transcripts were precipitated with 0.25 M NaCl in ethanol. After centrifugation, RNA pellets were dried and resuspended in autoclaved milli-Q water. The concentration of purified RNA samples was determined by absorbance measurement at 260 nm.

## In vitro translation assays in cell-free translation extracts

In vitro translation was carried out using increasing concentrations of mRNA transcripts with self-made untreated RRL, amino acid mixture containing all the amino acids except methionine (1 mM of each), RNasin (Promega ), 75 mM KCl, 0.5 mM MgCl$_2$, 3.8 mCi [$^{35}S$] methionine, and autoclaved milli-Q water. Reaction mixture was incubated at 30°C for 1 hr. Aliquots of translation mixture were analysed by SDS-PAGE (10%) (*Laemmli, 1970*) and translation products were visualized by phosphor imaging. In vitro translation assays with wheat germ extract (Promega) were performed according to manufacturer's instructions. In vitro translation assays with HeLa cell extract and *Drosophila* S2 cell extract were performed as previously described (*Thoma et al., 2004*; *Wakiyama et al., 2006*).

## Chemical probing

Probing experiments were performed as previously described (*Alghoul et al., 2021*).

### Probing with DMS

Modification by DMS was performed on 2 pmoles of each RNA (*Hoxa3* TIE and *Hoxa11* TIE). The RNA is first incubated for 15 min in DMS buffer (50 mM Na cacodylate [pH 7.5], 5 mM MgCl$_2$, and 100 mM KCl) and 1 µg of yeast total tRNA (Sigma- Aldrich) and then modified with 1.25% DMS reagent (diluted with ethanol 100%) with 10 min incubation at 20°C and stopped on ice. Modified transcripts were precipitated with 0.25 M NaCl, 0.1 mg/ml glycogen in ethanol. RNA pellets were dried and resuspended in autoclaved milli-Q water. Modified nucleotides were detected by primer extension arrests that were quantified. The intensity of the RT stops is proportional to the reactivity for each nucleotide.

### Probing with CMCT

Similarly, modification by CMCT was performed on 2 pmoles of each RNA (*Hoxa3* TIE and *Hoxa11* TIE). Each RNA is incubated for 20 min in CMCT buffer (50 mM Na borate [pH 8.5]; 5 mM MgCl$_2$; 100 mM KCl) and 1 µg of yeast total tRNA. Then modifications were performed with 10.5 g/l CMCT reagent with 20 min incubation at 20°C and stopped on ice. Modified transcripts were precipitated with 0.25 M NaCl, 0.1 mg/ml glycogen in ethanol. RNA pellets were dried and resuspended in auto-claved milli-Q water. Modified nucleotides were detected by primer extension arrests that were quantified. The intensity of the RT arrests is proportional to the reactivity for each nucleotide.

## Primer extension

RT was carried out in a 20 µl reaction with 2 pmoles of RNA and 0.9 pmoles of 5' fluorescently labelled primers. We used fluorescent Vic and Ned primers (Thermo Fischer Scientific) of same sequence for all RT reactions which are complementary to the *Hbb-b1* 5'UTR from nucleotides 6–37:
    5'-GGTTGCTAGTGAACACAGTTGTGTCAGAAGC-3'.

First, the RNA was unfolded by a denaturation step at 95°C for 2 min. Then, fluorescent primers were annealed for 2 min at 65°C followed by incubation on ice for 2 min. Samples were incubated in a buffer containing 83 mM KCl, 56 mM Tris-HCl (pH 8.3), 0.56 mM each of the four deoxynucleotides (dNTP), 5.6 mM DTT, and 3 mM MgCl$_2$. RT was performed with 1 unit of avian myoblastosis virus (AMV) reverse transcriptase (Promega) at 42°C for 2 min, 50°C for 30 min, and finally 65°C for 5 min. In parallel, sequencing reactions were performed in similar conditions, but supplemented with 0.5 mM dideoxythymidine or dideoxycitidine triphosphate (ddTTP or ddCTP) (protocol adapted from *Gross et al., 2017*). The synthesized cDNA were phenol–chloroform extracted and precipitated. After centrifugation, the cDNA pellets were washed, dried, and resuspended in 10 µl deionized Hi-Di formamide (freshly prepared highly deionized formamide). Samples were loaded on a 96-well plate for sequencing on an Applied Biosystems 3130xl genetic analyser. The resulting

electropherograms were analysed using QuSHAPE software (*Karabiber et al., 2013*), which aligns signal within and across capillaries, as well as to the dideoxy references of nucleotide at specific position and corrects for signal decay. Normalized reactivities range from 0 to 2, with 1.0–2.0 being the range of highly reactive positions. A preliminary secondary structure model was first initiated by mfold (*Mathews et al., 2016*) and then edited according to reactivity values.

## Sucrose gradient analysis

To analyse the assembly of ribosomal pre-initiation complexes on the RNA of interest, the complexes were loaded on 7–47% sucrose gradients containing 5 mM $MgCl_2$, 25 mM Tris-HCl (pH 7.5), 1 mM DTT, and 50 mM KCl. We used the specific translation inhibitors GMP-PNP (4 mM), cycloheximide (1 mg/ml), geneticin (0.7 mM), hygromycin (0.5 mg/ml), and edeine (10 mM), they were added to the RRL with a mix containing the 20 amino acids at 1.5 mM each, RNasin (Promega), 35 mM KCl, and 0.24 or 2.4 mM $MgCl_2$, prior to incubation with the 5' capped radioactive mRNA of interest. The assembled pre-initiation complexes were formed by incubation in RRL at 30°C for 5 min. Then, 8 mM $MgAc_2$ was added and one volume of 7% sucrose. Samples were then layered on the surface of 11 ml 7–47% sucrose gradient and centrifuged for 2 hr 30 min in an SW41 rotor at 37,000 rpm at 4°C. After centrifugation, the whole gradient was fractionated, and the mRNA was localized by measuring radioactivity in each collected fraction by Cerenkov counting in a scintillation counter.

## Mass spectrometry and data processing

Protein extracts were digested with sequencing-grade trypsin (Promega) as previously described (*Chicher et al., 2015*; *Prongidi-Fix et al., 2013*). Peptide digests were analysed by nano LC-MS/MS and MS data were searched by the Mascot algorithm against the UniProtKB database from *Oryctolagus cuniculus* (rabbit). Identifications were validated with a protein false discovery rate of less than 1% using a decoy database strategy. The total number of MS/MS fragmentation spectra was used to quantify each protein from three independent biological replicates. This spectral count was submitted to a negative -binomial test using an edge R GLM regression through the R-package. For each identified protein, an adjusted p-value corrected by Benjamini–Hochberg was calculated, as well as a protein fold change (FC is the ratio of the average of spectral counts from a specific complex divided by the average of spectral counts from a reference protein complex). The results were presented in a volcano plot using protein $\log_2$ FC and their corresponding adjusted $\log_{10}$p-values. The proteins that were up-regulated in each condition are shown in red (*Hoxa3* TIE versus *Hbb-b1* mRNA, *Hoxa11* TIE versus *Hbb-b1* mRNA, and *Hoxa3* TIE versus *Hoxa11* TIE).

## In vivo luciferase assay

For in vivo luciferase assay, HEK293T cells and C3H10T1/2 cells were transfected in six-well plates with various constructs of pmirGlo vector (Promega). Transfection was performed using Turbofect transfection reagent (Thermo Fischer Scientific) according to manufacturer's instructions. Cells were collected 24 hr post-transfection. Luciferase assay was performed using Dual-Glo luciferase kit (Promega) according to manufacturer's instructions. Firefly luciferase activities were measured to monitor transfection efficiency in order to normalize RLuc activities for each construct.

## Co-transfection assay of siRNAs and reporter plasmid in HEK293T cells

HEK293FT cells were used to test the effect on inhibition of eIF2D knock-down and of a non-target siRNA pool, as a negative control, on RLuc expression (*hRluc-neo*) in pmirGLO vectors. We used ON-TARGETplus human siRNAs against eIF2D (catalog number L-003680-01-00) and siRNA-NT (catalog number D-001810-10-05) purchased from Horizon Discovery. HEK293T cells were cultured according to manufacturer's instructions (ATCC) for 24 hr and used for co-transfection by siRNAs and reporter plasmid. We used $2 \times 10^5$ cells in 1 ml of culture medium without antibiotics. Upon reaching 70% confluence, cells were transfected by 5 pmoles of siRNAs in different wells with 500 ng of reporter plasmid. Transfections were performed with Lipofectamine 2000 (Invitrogen) according to manufacturer's instructions. After 48 hr, cells were washed twice by phosphate buffered saline (PBS $1\times$) and incubated with passive lysis buffer $1\times$ for 15 min. Luciferase assay was performed according to manufacturer's instructions. Protein concentration was measured by Bradford's assay.

The impact of silencing on TIE-mediated translation inhibition was measured by luciferase assay according to manufacturer's instructions as previously mentioned.

## Western blot against eIF2D and GAPDH

The silencing efficiency was quantified by western blots using rabbit polyclonal anti-eIF2D antibody (12840–1-AP) from Proteintech and mouse polyclonal anti-GAPDH antibody (sc-1377179) from Santa Cruz Biotechnology. For that, 20 µg of protein extracts were loaded on 10% polyacrylamide SDS-PAGE. After migration, proteins were transferred to an Immobilin-P membrane (Millipore) at 10 V for 1 hr in a semi-dry apparatus (Trans-Blot SD) on a PVDF (polyvinylidene fluoride) membrane that had been previously activated with 100% methanol for few seconds and a transfer buffer pH 8 (25 mM Tris; 200 mM glycine; 20% ethanol). After transfer, the membrane was saturated for 2 hr by blocking buffer (5% milk, 0.05% Tween-20, PBS 1×). Primary antibodies were added at dilutions recommended by the manufacturers in blocking buffer, the membranes were incubated overnight at 4°C. Then, the membranes were washed three times by PBST (PBS 1×; 0.05% Tween-20) to remove the excess of primary antibodies. Then, membranes were incubated with secondary HRP-conjugated antibody for 1 hr at room temperature followed by three washing steps. The signal produced by reaction between HRP and ECL (Kit ECL Plus Western Blotting Detection System, GE Healthcare) was detected by chemiluminescence using imaging Chemidoc (Biorad).

## Ribosome toe printing assay

Toe printing assay was adapted from previously established protocols (*Martin et al., 2016*; *Martin et al., 2011*). Briefly, RRLs were incubated for 5 min at 30°C then 10 min on ice with buffer containing 1 U/µl of recombinant RNasin (Promega), 75 mM KCl 0.5 mM $MgCl_2$, and 1.3 mM of puromycin prior to initiation complex assembly. Then, the pre-initiation complexes were formed by incubation with 500 nM of the RNA of interest in the presence of specific inhibitors such as cycloheximide (1 mg/ml) or GMP-PNP (4 mM) for 5 min at 30°C and then 20 min on ice. Then, the pre-initiation complexes were complemented with one volume of ice-cold buffer A containing 20 mM Tris-HCl (pH 7.5), 100 mM KAc, 2.5 mM $MgAc_2$, 2 mM DTT, 1 mM ATP, and 0.25 mM spermidine and placed on ice. In order to separate ribosomal complexes from the non-ribosomal fraction, samples were centrifuged at 88,000 rpm in S100AT3 rotor (Sorvall-Hitachi) at 4°C for 1 hr. After centrifugation, the pellets containing the pre-initiation complexes were resuspended in 30 µl of ice-cold buffer A and incubated with 5' radioactively labelled DNA oligonucleotide complementary to nts 22–51 of RLuc coding sequence for 3 min at 30°C. Then, 1 µl of a 320 mM Mg(Ac)$_2$, 4 µl of a dNTP mixture (containing 5 mM of dATP, dGTP, dTTP, and dCTP), 10 units of recombinant RNAsin (Promega), and 1 unit/µl AMV reverse transcriptase (Promega) were added and incubated for 1 hr at 30°C. The synthesized cDNAs were analysed on 8% PAGE next to sequencing ladders.

## Acknowledgements

This work was funded by 'Agence Nationale pour la Recherche' (Ribofluidix, ANR-17-CE12-0025-01), by University of Strasbourg, and by the 'Centre National de la Recherche Scientifique'. We would like to thank Dr Maria Barna for sharing sequences of *Hoxa3* and *Hox11* transcripts. We are grateful to Dr Christine Allmang, Hassan Hayek, Antonin Tidu, Aurélie Janvier, Aurélie Durand for technical assistance, Philippe Hamman, Johana Chicher, Lauriane Kuhn for mass spectrometry analysis, Dr Mireille Baltzinger and Dr Pascale Romby for support and useful discussions on the project. We would also like to thank Dr Sebastien Pfeffer for the pmirGLO plasmid.

## Additional information

### Funding

| Funder | Grant reference number | Author |
|---|---|---|
| Agence Nationale de la Recherche | ANR-17-CE12-0025-01 | Franck Martin |

The funders had no role in study design, data collection and interpretation, or the decision to submit the work for publication.

### Author contributions
Fatima Alghoul, Conceptualization, Formal analysis, Investigation, Methodology, Writing - original draft; Schaeffer Laure, Formal analysis, Investigation; Gilbert Eriani, Conceptualization, Formal analysis, Funding acquisition, Methodology, Writing - original draft; Franck Martin, Conceptualization, Formal analysis, Supervision, Funding acquisition, Investigation, Methodology, Writing - original draft, Project administration

### Author ORCIDs
Fatima Alghoul (iD) https://orcid.org/0000-0003-1934-9483
Franck Martin (iD) https://orcid.org/0000-0001-9724-4025

### Decision letter and Author response
Decision letter https://doi.org/10.7554/eLife.66369.sa1
Author response https://doi.org/10.7554/eLife.66369.sa2

## Additional files

### Supplementary files
• Supplementary file 1. Mass spectrometry analysis of pre-initiation complexes programmed with *Hoxa3* TIE, *Hoxa11* TIE, and 5'UTR *Hbb-b1* in the presence of cycloheximide.

• Supplementary file 2. Mass spectrometry analysis of pre-initiation complexes programmed with *Hoxa3* TIE, *Hoxa11* TIE, and 5'UTR *Hbb-b1* in the presence of GMP-PNP.

• Transparent reporting form

### Data availability
All data generated or analysed during this study are included in the manuscript and supporting files. Source data files have been deposited on DRYAD: DOI https://doi.org/10.5061/dryad.j3tx95xdg.

The following dataset was generated:

| Author(s) | Year | Dataset title | Dataset URL | Database and Identifier |
|---|---|---|---|---|
| Alghoul F, Schaeffer L, Eriani G, Martin F | 2021 | Translation Inhibitory Elements from Hoxa3 and Hoxa11 mRNAs use uORFs for translation inhibition | http://dx.doi.org/10.5061/dryad.j3tx95xdg | Dryad Digital Repository, 10.5061/dryad.j3tx95xdg |

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
