## [Decision Letter]

**Acceptance summary:**

Homeobox (Hox) genes encode many mRNAs whose expression is controlled at the level of translation via translation inhibitory elements (TIE) that impair cap-dependent translation. The paper reports that the translation of Hox a3 and a11 is regulated by different mechanisms. The inhibitory mechanism that is mediated by TIE a3 requires the translation initiation factor eIF2D, while The inhibitory mechanism of TIE a11 is mediated by three elements: an upstream start codon (uAUG), followed by a stop codon, and a long stable hairpin. These findings show that the TIE elements of Hox mRNAs control translation via novel mechanisms.

**Decision letter after peer review:**

Thank you for submitting your article "Translation Inhibitory Elements from Hox a3 and a11 mRNAs use uORFs for translation inhibition" for consideration by *eLife*. Your article has been reviewed by 3 peer reviewers, and the evaluation has been overseen by a Reviewing Editor and James Manley as the Senior Editor. The reviewers have opted to remain anonymous.

Given the importance of HoxA mRNAs in development the paper is of interest. The study is comprehensive and thorough. A revised version needs to address the listed below criticisms and concerns. The full text of the reviewers' comments is appended, inasmuch as they contain many additional thoughtful and constructive criticisms.

1. The mass spectral tables must be shown. Also, some validation of at least some of the data is necessary (reviewer #2).

2. The ribosome binding assays are problematic and need to be rectified (reviewer #2). Reviewer #1 raised issues with peak/fraction assignments.

3. Figure 7A (reviewer #1). It is not clear why is it expected that reduction in eIF4E levels would interfere with TIE-mediated inhibition on translation. Also, internal controls, which are known to be (or not be) responsive to eIF4E and eIF2D suppression are lacking making it difficult to judge whether the changes in expression are specific.

4. It is not clear why specifically Hox a3 and a11 were investigated, and not the other HoxA mRNAs (reviewer #3).

5. Figure 6 (reviewer#3), both TIE a3 and a11 are showing RPL38 ribosomal protein. Why is it stated on page 38 that Hox a3 mRNA is RPL38-independent?

6. In vitro and cell-based experiments assessing the translational activity of reporters lack internal reference controls. What is usually used in these types of experiments is an IRES-driven reporter that can inform on non-specific inhibition events and control for transfection efficiencies

7. The paper is too long.

*Reviewer #1 (Recommendations for the authors):*

There are many instances (in other contexts) where uORFs or secondary structure within 5' UTRs have been found to be detrimental to translation initiation. Whereas the current study provides insight into the specific regulation of the Hox a3 and Hox a11 mRNAs, the mechanism elucidated is not particularly novel.

1. In vitro and cell-based experiments assessing the translational activity of reporters lack internal reference controls (unless I am missing something on how the data was processed). What is usually used in these types of experiments is an IRES-driven reporter that can inform on non-specific inhibition events and control for transfection efficiencies. As well, the authors have not ruled out differential mRNA stabilities contributing to reduced output from test mRNAs in any of their experiments.

2. FiguresS3B, 5C and 8 – In this ribosome binding assay, we do not see the distribution of the ribosomes throughout the gradient, only the radiolabelled counts of the mRNA are plotted as % of input. How do the authors know to assign polysomes, 80S, 48S, RNP to specific parts of the gradient? I'm assuming they are using the migration of their labelled mRNA in the presence of drugs whose mechanism of action are known to infer there the polysomes, 80S, 48S, RNP are located, but this is indirect. Normally, when these polysome binding studies are understaken, one monitors the OD260 profile to directly observe the ribosome distribution throughout the gradient. The mRNA counts are then layered on top of this.

3. The authors undertake Mass Spect analysis of pre-initiation complexes purified by affinity chromatography and report differences in protein-RNA complexes when complexes were formed on TIE a3 versus TIE a11 globin mRNA. I couldnt really get a sense of how well the purification worked, on the quality of the MS data, nor could I find comprehensive experimental validation performed on several of the "hits". The authors refer to eIF2D (as well as eIF1A, eIF1a-X, eIF5B, LARS, QARS, DHX21, HBS1L [p. 16]) as TIE a3 enriched factors and make the statement that this hints at two distinct TIE-mediated inhibitions. I disagree with this statement since factors, such as eIF5B, eIF1A, LARS, QARS, are required for the basic translation process and their presence will simply reflect which stage of translation is captured/enriched in the affinity chromatography experiments. I think this preliminary data is over-interpreted in its current form. Since PKR appears in the a3 and a11 complexes, is eIF2alpha phosphorylated in cells expressing these reporters?

4. The labelling on siRNA-eIF2D panel for TIE a3 in Figure 7 should be better positioned above the panel.

5. I don't understand the results with si4E in Figure 7A. Why is it expected that reduction in eIF4E levels would interfere with TIE-mediated inhibition on translation? As well, internal controls known to be (or not be) responsive to eIF4E and eIF2D suppression are lacking here in order to be able to know if changes in expression are specific.

6. I'm not sure the cycloheximide experiment in Figure 8 adds to the understanding of the a3 TIE. It's not that complexes are "resistant" to cycloheximide (as the authors state in their subtitle), it's just that different complexes are observed compared to the reporter without the TIE 5' UTR. Without knowing the basis for these additional complexes, or their composition, this experiment is of limited value since a clear interpretation is difficult to make.

*Reviewer #2 (Recommendations for the authors):*

1. The tables for the mass spectrometry experiments were not included. These should be added.

2. Based on the data presented, it seems the authors are overemphasizing the "start-stop" part of the mechanism for a11. A better wording would note the generality of the SL for a11, and a species-specific use of a "start-stop" uORF upstream of the SL.

3. All of the data points for the bar graphs should be plotted on top of the bars.

*Reviewer #3 (Recommendations for the authors):*

1. There are 9 figures in the manuscript that should be consolidated, as it is cumbersome for the reader to follow.

2. What is the relation between Hox and the *Drosophila* genes Antp and Ubx? This is not clear in the Introduction section.

3. It is not clear why specifically Hox a3 and a11 were investigated, and not the other HoxA mRNAs.

4. The grammar of manuscript needs correction, and many sentences are missing appropriate punctuation which adds to confusion

5. The term in vivo refers to research done with or within an entire living organism. The term in vitro refers to research done outside of a living organism, this included studying cells in culture. The term in vivo should be corrected in the manuscript. OR better yet, don't use either term because they are context dependent.

6. The mutations performed in AUG123 from Figure S3A (right panel) are not described in the manuscript.

7. What does FLa3 stand for in Figure 4A?

8. Why does Figure S4B (right panel) have two lanes for Hox a3 and w/o TIE? Are they representing duplicate results? The lanes should be labeled.

9. Are the results from Figure S5 statistically significant?

10. Figure 6 (middle panel) is not showing QARS, as mentioned in the manuscript on page 16, line 8.

11. Figure 6, both TIE a3 and a11 are showing RPL38 ribosomal protein. Why does page 38 states that Hox a3 mRNA is RPL38-independent?

12. Page 17 lines 1-5, the percentage mentioned of 74% and 76% are respectively to Hox a3 and a11? This should be clarified.

13. The silencing of eIF2D at efficiency of 81% has no significant effect on a11-mediated inhibition compared to which sample? Needs explanation.

14. The text describing GMP-PNP and cycloheximide from page 18 should be moved to the first mention of each inhibitor in the manuscript.

15. In Figure 7, the blots should be improved and experiments repeated in the 10T_1/2_ cells (that are used elsewhere in the manuscript) which mimic the embryonic development and differentiation.

16. One of the major weaknesses is that there is no functional assay to really assess the role of these findings.

---

## [Author Response]

Given the importance of HoxA mRNAs in development the paper is of interest. The study is comprehensive and thorough. A revised version needs to address the listed below criticisms and concerns. The full text of the reviewers' comments is appended, inasmuch as they contain many additional thoughtful and constructive criticisms.1. The mass spectral tables must be shown. Also, some validation of at least some of the data is necessary (reviewer #2).

The mass spectrometry data on complexes purified after GMP-PNP and cylcoheximide treatments are now presented as Excel files in supplementary materials (Supplementary tables 1 and 2). As presented in the manuscript, we have validated the involvement of eIF2D in TIE a3-mediated inhibition in vivo by an RNA interference approach. Importantly, eIF2D does not seem to be involved in TIE a11-mediated inhibition. Since our mass spectrometry analysis revealed that eIF2D is found only with a3 but not with a11 (Figure 6), our functional data (New Figure 7) are in agreement with our Mass Spectrometry data.

2. The ribosome binding assays are problematic and need to be rectified (reviewer #2). Reviewer #1 raised issues with peak/fraction assignments.

We acknowledge the reviewers for their constructive comments. In these experiments, we incubate 5’-capped and radiolabeled mRNA transcripts in rabbit reticulocyte lysates. The coloured sections corresponding to polysomes, 80S, 48S and RNP have been assigned according to the OD profiles at 254 nm for each sucrose gradient fractionation. To improve the clarity of the profiles, we have chosen not to superimpose all the OD profiles on top of the radioactivity curves. However, we have generated a new supplemental figure S6 showing an example of the OD profile on top of the radioactivity curves to clarify how the coloured sections have been assigned. We have also explained in more details the experiments and the coloured section assignment in figure 5C legend.

3. Figure 7A (reviewer #1). It is not clear why is it expected that reduction in eIF4E levels would interfere with TIE-mediated inhibition on translation. Also, internal controls, which are known to be (or not be) responsive to eIF4E and eIF2D suppression are lacking making it difficult to judge whether the changes in expression are specific.

The confusion comes from the fact that siRNA against eIF4E have additive effects on two mechanisms, first it affects TIE-mediated inhibition since TIE are using uORF that are translated by a cap-dependent mechanism but the siRNA against eIF4E also impacts the expression of the main ORF, the Renilla coding sequence (RL) which is translated by leaky scanning of the uORF and, more importantly the second reporter Firefly luciferase (FL) that is used for normalization. Therefore, the reviewer is right, the conception of this experiment is not appropriate. We thank the reviewer for pointing this contradiction. Indeed, we cannot use the siRNA against eIF4E in this experiment. First, we improved the presentation of the experimental procedure in figure 7A. We clearly show that the plasmid pmirGLO used harbours two reporter genes, Renilla luciferase to test the impact of TIEs and Firefly luciferase for normalization purposes. We also remodelled the presentation of this experiment and limited it to the assessment of the impact of siRNA against eIF2D only. Our current figure shows the RL/FL ratio for each co-transfection experiment. Using this ratio, we can compare the effect of silencing eIF2D to that of nontarget siRNAs in a3, a11 and pmirGLO constructs. In this latter case, the design of this experiment is appropriate since eIF2D silencing does not impact the translation of the main ORF. This is corroborated by the fact that eIF2D silencing has no impact on the second internal control reporter, Firefly luciferase. Another important result of this experiment is that silencing of eIF2D impacts specifically a3 but not a11. This also confirms our Mass Spec data that demonstrates that eIF2D is recruited specifically on a3 but not on a11. The figure 7A legend has been corrected accordingly.

4. It is not clear why specifically Hox a3 and a11 were investigated, and not the other HoxA mRNAs (reviewer #3).

We choose to work with a3 and a11 for several reasons. First, among hox, the translation mechanisms used are distinct (Xue et al., 2015). In fact some hoxA mRNAs are translated by an RPL38-mediated mechanism whereas some others do not require RPL38 to be present on the ribosome, which hints at least two different mechanisms. We wanted to investigate further an example of each of these two categories; Hox a11 mRNA translation being RPL38-dependent while Hox a3 is not. Another important criteria for the choice of a3 and a11 is their high inhibition rate of their TIE element, a3 and a11 are among the best (Xue et al., 2015). Last but not least, we chose a3 and a11 because of their small 5’UTR size, which makes them more suitable for several experiments that were conducted in our study, which required pure RNA transcripts with a high purification quality. Indeed, such experiments are more challenging with longer transcripts. These three arguments are now explained in the introduction of the revised manuscript.

5. Figure 6 (reviewer#3), both TIE a3 and a11 are showing RPL38 ribosomal protein. Why is it stated on page 38 that Hox a3 mRNA is RPL38-independent?

The reviewer is right, our mass spectrometry data showed that pre-initiation complexes assembled with both TIE elements contain RPL38. This seems to be contradictory because in the literature, only hoxa11 translation has been described to be RPL38independent (Xue et al., 2015). In fact, all the ribosomal proteins are present in these complexes, which is expected since we purified a full ribosome and we don’t expect specific depletion of RPL38 from the ribosome. This does not mean that RPL38 is needed for translation but it is present in our analysis just because it is part of the ribosome. However, the reviewer is right our statistical analysis shows that RPL38 is enriched with both a3 and a11 TIEs. Although this is interesting, so far, we have no rationale to explain this fact, we added two sentences to the manuscript to discuss this point in the result section. We also want to mention that the study from Xue et al. addresses the role of RPL38 for IRES functioning yet no study has been conducted to address the role of RPL38 for TIE functioning.

6. In vitro and cell-based experiments assessing the translational activity of reporters lack internal reference controls. What is usually used in these types of experiments is an IRES-driven reporter that can inform on non-specific inhibition events and control for transfection efficiencies

The reviewer is right, such experiments need internal controls. Concerning in vitro cellfree experiments, we did perform quality checks of our extracts using various reporter genes. Using a standard reporter construct containing the β-globin 5’UTR upstream of a Renilla coding sequence, we showed that translation is predominantly cap-dependent as expected in our self-made extracts from rabbit reticulocyte lysate. We have included in supplementary figure 1A, an additional panel that shows cap-dependency of RRL with a reporter mRNA without TIE. Concerning the in vivo experiments, we have in fact included an internal control in all our experiments. The plasmid used harbours two reporter genes, a Renilla luciferase in which we test the inhibition of TIEs and a Firefly luciferase that is translated in a canonical cap-dependent manner. All the results have been normalized against this internal control. In the revised version of the manuscript, we have clarified this point by inserting a new supplementary figure S1A for the in vitro experiments reference controls that we have used for assessing the translational activity of our cell-free extracts.

7. The paper is too long.

We have removed figure 8 and the discussion about this point to shorten the manuscript.

Reviewer #1 (Recommendations for the authors):There are many instances (in other contexts) where uORFs or secondary structure within 5' UTRs have been found to be detrimental to translation initiation. Whereas the current study provides insight into the specific regulation of the Hox a3 and Hox a11 mRNAs, the mechanism elucidated is not particularly novel.1. In vitro and cell-based experiments assessing the translational activity of reporters lack internal reference controls (unless I am missing something on how the data was processed). What is usually used in these types of experiments is an IRES-driven reporter that can inform on non-specific inhibition events and control for transfection efficiencies. As well, the authors have not ruled out differential mRNA stabilities contributing to reduced output from test mRNAs in any of their experiments.

Concerning RNA controls, we have answered to this point (see point 6 of general comments). For the stability of our transcripts in cell-free translation extracts, we want to emphasize that we are using capped-mRNA transcripts in addition to RNAse inhibitors. The use of these cell-free extracts has been assessed carefully in our lab since several years. In particular, we have shown previously that there is no RNA degradation is these assays (Martin et al., 2011; Mol Cell, Martin et al., 2016 Nat Comm, Gross et al., 2018 NAR, Martin et al., 2018 Nat Comm). Concerning in vivo assays, we used the plasmid pmirGlo that contains two reporters. This allows internal standardization of each assay by normalization of the same reporter in every assay. Figure 7 has been clarified to better explain the experimental procedure of our in vivo assay.

2. Figures S3B, 5C and 8 – In this ribosome binding assay, we do not see the distribution of the ribosomes throughout the gradient, only the radiolabelled counts of the mRNA are plotted as % of input. How do the authors know to assign polysomes, 80S, 48S, RNP to specific parts of the gradient? I'm assuming they are using the migration of their labelled mRNA in the presence of drugs whose mechanism of action are known to infer there the polysomes, 80S, 48S, RNP are located, but this is indirect. Normally, when these polysome binding studies are undertaken, one monitors the OD260 profile to directly observe the ribosome distribution throughout the gradient. The mRNA counts are then layered on top of this.

We acknowledge the reviewer for these constructive comments. Yes, we haven’t shown the OD profiles in purpose. We didn’t want to superimpose all the OD profiles to improve the clarity of our plots. However, we have assigned the polysomes, 80S, 48S and RNP sections according to the OD profiles. As an example, we inserted an additional supplementary figure S6 showing one OD profile on top of the sucrose gradient from figure 5. We also implemented the figures 5C and S6 legends with a better description of the assignments of each section with the OD profile. In conclusion, we believe that overlaying OD profiles on all the plots will be detrimental to the clarity of the figures and we would prefer not to show them but rather keep the coloured sections.

3. The authors undertake Mass Spect analysis of pre-initiation complexes purified by affinity chromatography and report differences in protein-RNA complexes when complexes were formed on TIE a3 versus TIE a11 globin mRNA. I couldn’t really get a sense of how well the purification worked, on the quality of the MS data, nor could I find comprehensive experimental validation performed on several of the "hits". The authors refer to eIF2D (as well as eIF1A, eIF1a-X, eIF5B, LARS, QARS, DHX21, HBS1L [p. 16]) as TIE a3 enriched factors and make the statement that this hints at two distinct TIE-mediated inhibitions. I disagree with this statement since factors, such as eIF5B, eIF1A, LARS, QARS, are required for the basic translation process and their presence will simply reflect which stage of translation is captured/enriched in the affinity chromatography experiments. I think this preliminary data is over-interpreted in its current form. Since PKR appears in the a3 and a11 complexes, is eIF2alpha phosphorylated in cells expressing these reporters?

In the revised version of the manuscript, we included the results of our Mass Spec analysis. We would like to emphasize that the complex purifications have been done in three independent experiments. Analysis of the multidimensional scaling plots, which illustrate the global variance and similarities between the GMP-PNP complexes and cycloheximide complexes detected in the replicates, after a median-to-ratio normalization, shows that our purification procedure is robust and highly reproducible. Moreover, we have used routinely this purification procedure for pre-initiation complexes isolation (Prongidi et al., 2013; Chicher et al., 2015) that allowed us to faithfully characterizer specific eIFs on CrPV IRES (Gross et al., 2017, NAR) and pre-initiation complexes assembled on histone H4 mRNA with a high purity grad that allowed further characterization and visualization by Cryo-EM (Martin et al., 2016, Nat Comm). In the volcano plots presented in the manuscript, we show only the differences between complexes. We now present as supplementary material the full datasets as supplementary tables 1 and 2. The outcome of our experiments clearly shows that we purified pre-initiation complexes that all contain ribosomal proteins and eukaryotic initiation factors indicating functional complexes.

Among specific proteins, the reviewer is right that there are other very interesting candidates. We chose to focus on eIF2D in this manuscript for further functional validation. Nevertheless, other candidates will be pursued in future studies. We added a comment in the results and in the Discussion sections in the revised manuscript on two interesting candidates, eIF5B and HBS1L as pointed out by reviewer #2. Concerning PKR, the reviewer points out an interesting question, we have not checked whether eIF2 is phosphorylated in cells expressing TIE reporters, we will also address this issue in future investigations.

4. The labelling on siRNA-eIF2D panel for TIE a3 in Figure 7 should be better positioned above the panel.

Corrected.

5. I don't understand the results with si4E in Figure 7A. Why is it expected that reduction in eIF4E levels would interfere with TIE-mediated inhibition on translation? As well, internal controls known to be (or not be) responsive to eIF4E and eIF2D suppression are lacking here in order to be able to know if changes in expression are specific.

Figure 7 has been totally remodelled in order to address the concerns from the reviewer. (see point 3 of general comments).

6. I'm not sure the cycloheximide experiment in Figure 8 adds to the understanding of the a3 TIE. It's not that complexes are "resistant" to cycloheximide (as the authors state in their subtitle), it's just that different complexes are observed compared to the reporter without the TIE 5' UTR. Without knowing the basis for these additional complexes, or their composition, this experiment is of limited value since a clear interpretation is difficult to make.

We are grateful to the reviewer for this comment. We agree that this experiment is not conclusive and although we are convinced that the unconventional profile that was observed in the presence of cycloheximide is interesting, we don’t have a rational explanation so far. Since the paper is too long (point 7), we propose to remove Figure 8 from the manuscript and the comments related to this figure (i.e. point 6 of general comments).

Reviewer #2 (Recommendations for the authors):1. The tables for the mass spectrometry experiments were not included. These should be added.

Mass spectrometry data are presented as supplementary tables 1 and 2.

2. Based on the data presented, it seems the authors are overemphasizing the "start-stop" part of the mechanism for a11. A better wording would note the generality of the SL for a11, and a species-specific use of a "start-stop" uORF upstream of the SL.

We acknowledge this constructive comment, we have modified the manuscript accordingly.

3. All of the data points for the bar graphs should be plotted on top of the bars.

All the data presented in the histograms are the average of three independent replicas. Since the error bars are generally very small in all the presented histograms, we don’t believe that the requested modification is necessary.

Reviewer #3 (Recommendations for the authors):1. There are 9 figures in the manuscript that should be consolidated, as it is cumbersome for the reader to follow.

As previously mentioned, we have removed figure 8.

2. What is the relation between Hox and the *Drosophila* genes Antp and Ubx? This is not clear in the Introduction section.

Antp and Ubx genes belong to *Drosophila* Hom-C cluster genes, which are the homologues of Hox genes in vertebrates. It has been shown that there exists a level of translational control due to cap-independent activity by IRES elements in these mRNAs that plays a role in development. According to Xue et al., a similar IRES-mediated mechanism controls the translation of HoxA mRNAs during embryonic development. The introduction section has been modified accordingly.

3. It is not clear why specifically Hox a3 and a11 were investigated, and not the other HoxA mRNAs.

See point 4.

4. The grammar of manuscript needs correction, and many sentences are missing appropriate punctuation which adds to confusion

Corrected.

5. The term in vivo refers to research done with or within an entire living organism. The term in vitro refers to research done outside of a living organism, this included studying cells in culture. The term in vivo should be corrected in the manuscript. OR better yet, don't use either term because they are context dependent.

The reviewer is right however we are not working on animal models and we find it is better to differentiate our in-vitro experiments performed in tubes from our cell line based experiments. So, we would prefer to keep the term in vitro for cell-free translation experiments and in vivo for those performed in cell lines.

6. The mutations performed in AUG123 from Figure S3A (right panel) are not described in the manuscript.

Corrected.

7. What does FLa3 stand for in Figure 4A?

FLa3 refers to full length 5’UTR of Hox a3 which constitutes the TIE and IRES elements. We added this detail to the figure legend.

8. Why does Figure S4B (right panel) have two lanes for Hox a3 and w/o TIE? Are they representing duplicate results? The lanes should be labeled.

These are duplicate experiments. We mentioned this in the figure legend.

9. Are the results from Figure S5 statistically significant?

Yes, the results are statistically significant. We modified the figure by adding the P values represented by the asterisks.

10. Figure 6 (middle panel) is not showing QARS, as mentioned in the manuscript on page 16, line 8.

We have modified the text in the manuscript accordingly.

11. Figure 6, both TIE a3 and a11 are showing RPL38 ribosomal protein. Why does page 38 states that Hox a3 mRNA is RPL38-independent?

This question was addressed in point #5 in the reviewing editor’s comments.

12. Page 17 lines 1-5, the percentage mentioned of 74% and 76% are respectively to Hox a3 and a11? This should be clarified.

Corrected.

13. The silencing of eIF2D at efficiency of 81% has no significant effect on a11-mediated inhibition compared to which sample? Needs explanation.

We have remodelled our figure (new figure 7) and presented a better analysis for the results. Our conclusion was based on the comparison of RL/FL values in eIF2D silencing and in the non-target pool of siRNAs. The difference between the two RL/FL values would represent the effect of eIF2D silencing on a11-mediated inhibition. Accordingly, RL/FL values upon eIF2D silencing (0.017) and upon using non-target pool of siRNAs (0.02) are almost the same then we can conclude that there is no significant effect of eIF2D silencing on a11-mediated inhibition.

14. The text describing GMP-PNP and cycloheximide from page 18 should be moved to the first mention of each inhibitor in the manuscript.

Corrected. Description of cycloheximide and GMP-PNP was moved to page 13 in the manuscript.

15. In Figure 7, the blots should be improved and experiments repeated in the 10T_1/2_ cells (that are used elsewhere in the manuscript) which mimic the embryonic development and differentiation.

HEK293 cell line is an embryonic cell line which expresses Hox transcript. Our protocols were optimized according to this cell line as transfection was more efficient in this cell line. Futures studies will focus on improving these experiments in 10T_1/2_ cell line.

16. One of the major weaknesses is that there is no functional assay to really assess the role of these findings.

We agree with this point. However, following hox expression in vivo is very challenging, it is well described that western blot for Hox protein are very difficult because reliable antibodies are not easily available. According to (Luo et al.,2019: PMID: 30866492), one of the main technological challenges to study Hox is that: “Most commercial antibodies to the HOX proteins are of very low quality (perhaps due to the strong evolutionary conservation of their protein sequences) and cannot even detect exogenously expressed high levels of the target HOX protein”.